# ERBB2 signaling drives immune cell evasion and resistance against immunotherapy in small cell lung cancer

Lydia Meder [1,2,20] ✉, Charlotte I. Orschel[3,4,5,20], Cyrielle L. Bouchez[3,4,5,20], Rahil Gholamipoorfard[3,6], Claudia V. Orschel[3,5], David Stahl[3,4,5], Christoph Kreer [7], Mirjam Koker[3,4], Marieke Nill[3,4], Ilayda G. Kocak[1], Ka-Won Noh[8], Xinlei Zhao[8], Leon Ullrich [7], Björn Häupl[9,10,11,12], Josefine Jakob[9,10,11], Marie-Lisa Eich[8,13], Alexandra Florin[8], Holger Grüll[14,15], Jürgen Wolf[3,16], Filippo Beleggia [3,5,6], Reinhard Büttner [4,8,16], Thomas Oellerich[9,10,11,12], Florian Klein [7], Johannes Brägelmann [4,5,6,17], H. Christian Reinhardt [18], Nima Abedpour [3,6,19] & Roland T. Ullrich[3,4,5,16] ✉

Small cell lung cancer (SCLC) is characterized by its highly aggressive phenotype and dismal outcome. Despite the benefit of adding immune checkpoint blockade to standard chemotherapy, tumors acquire the ability to evade immunosurveillance and develop resistance. To investigate these underlying mechanisms, we perform high-dimensional profiling of human and murine SCLC specimens. In matched primary and metastatic human samples, we observe MHC-I loss in metastases, highlighting its role in immune evasion. Correspondingly, silencing MHC-I in SCLC cells drastically reduces immune infiltration and promotes metastasis in mice. Using mass spectrometry and phospho-tyrosine kinase analyses, we identify ERBB2 signaling as a suppressor of MHC-I and driver of immune-modulatory transcripts. Mechanistically, genetic and pharmacologic blockade of ERBB2 induces MHC-I in a STING-dependent manner and prevents immune evasion in autochthonous murine SCLC. Strikingly, combining ERBB2 inhibition with anti-PD-1 elicits profound synergistic responses in preclinical models, suggesting this combination for future clinical trials in SCLC patients.

Small cell lung cancer (SCLC) represents about 15% of primary lung cancers and is characterized by obligate bi-allelic mutations with a loss of *RB1* and *TP53*. In contrast to non-small cell lung cancer (NSCLC), accounting for about 85% of lung cancer patients, targetable oncogenic drivers are uncommon in SCLC[1,2]. Patients with SCLC show remarkable responses to initial standard chemotherapy. Nevertheless, SCLC usually relapses after only a few months, resulting in dismal overall survival of about 10 months[3]. A recent clinical study (IMpower133) demonstrated improved overall survival by the addition of anti-PD-L1 targeted immune checkpoint blockade.

However, despite improved treatment response, patients succumbed after 12.3 months of therapy[3,4]. Metastatic spread is one of the strongest parameters that is associated with a dismal outcome in SCLC patients[3]. Most patients are diagnosed at advanced stages with liver or brain metastases[5]. The exact mechanism being responsible for this immune-evasive, metastatic behavior remains largely unknown[6]. Thus, there is a considerable unmet need to decipher targetable mechanisms that regulate tumor immune evasion and metastasis to develop new treatment approaches to improve outcome in patients with SCLC.

Immune checkpoint therapies against PD-1 or CTLA-4 have shown that an already present, endogenous immune response can potentially regress human tumors, if reactivated. This effective anti-tumor T cell response upon immune checkpoint blockade was first described in patients with relapsed metastatic melanoma treated with an antibody targeting CTLA-4[7]. However, long-lasting responses upon immune checkpoint inhibition are uncommon in patients with SCLC[3,4]. The key effectors of tumor elimination upon immune checkpoint blockade are CD8+ cytotoxic T cells that recognize for-eign antigens bound to major histocompatibility class I (MHC-I) molecules[8]. In line with these findings, neuroendocrine SCLC with high T-effector signals demonstrated longer overall survival with PD-L1 blockade combined with chemotherapy[9].

In this work, we elucidate the impact of loss of MHC-I expression upon ERBB2 signaling on tumor immune cell evasion and tumor metastasis in SCLC. Our data provide a mechanistic insight for how SCLC loses MHC-I expression enabling immune cell evasion and strong evidence that combined inhibition of ERBB2 and PD-1 improves out-comes for SCLC patients.

## Results

### Loss of MHC-I mediates immune cell evasion and triggers the formation of metastases in SCLC

In order to decipher the molecular patterns that are associated with immune cell evasion and metastasis in SCLC, we analyzed publicly available single-cell RNA sequencing (scRNA-seq) data[10] and sear-ched for differentially expressed genes between primary and metastatic lesions. Here, we observed a significant downregulation of *B2M* in metastases compared to primary tumor lesions of patients with SCLC (Fig. 1a–c). *B2M* encodes β2-microglobulin, a non-polymorphic subunit of the MHC-I complex essential for its sur-face expression and stability[11]. To confirm our transcriptional find-ings, we further examined MHC-I protein expression via immunohistochemistry in a unique collection of six matched sam-ples of primary and metastatic SCLC lesions (Fig. 1d). We found that MHC-I expression is strongly reduced in liver metastases in direct comparison to matched primary SCLC in the lung of the same patients (Fig. 1d, e). We next aimed to validate this finding in an autochthonous, conditionally *Rb1/Trp53*-depleted SCLC mouse model and again analyzed matched samples of primary and meta-static SCLC lesions. In line with our SCLC patient-derived data, we found that MHC-I expression was strongly reduced in metastatic lesions, compared to the primary SCLC tumor (Fig. 1f). We thus speculated that loss of MHC-I mediates immune evasion in meta-static SCLC. We next sought to investigate the impact of loss of MHC-I on SCLC cells in an immunocompetent mouse model. Using CRISPR-CAS9 KO technology, we successfully knocked out MHC-I on murine SCLC cells (Fig. 2a–c). We injected these SCLC MHC-I KO intravenously into immunocompetent and immunodeficient mice and monitored the formation of liver metastases by MRI (Fig. 2d). Strikingly, KO of MHC-I on SCLC cells abrogated immuno-surveillance in an immunocompetent model and induced a massive metastatic spread in the liver and lymph nodes (Fig. 2e) accom-panied by T cell reduction and reduced CD8 T cell activity (Fig. 2f, g). At the same time, we did not observe an increased infiltration of activated NK cells in MHC-I KO tumors (Supplementary Fig. 1a). Further metastatic spread was limited due to rapid massive spread into the liver and early death (Supplementary Fig. 1b). In contrast, intravenous injection of WT and MHC-I KO led to an equivalent metastatic engraftment in immunodeficient mice (Fig. 2h). Fur-thermore, we injected SCLC MHC-I KO or SCLC WT orthotopically into immunocompetent mice and observed that MHC-I KO cells generated a higher tumor load in the lung than SCLC WT cells and were characterized by a markedly reduced CD8 T cell infiltration, indicating that loss of MHC-I evades T cell immunosurveillance

(Supplementary Fig. 1c–k). Thus, our data strongly indicates that loss of MHC-I on SCLC cells drives immune cell evasion and the formation of metastases in SCLC.

### ERBB2 regulates the MHC-I antigen presentation pathway and inhibition of ERBB2 enhances MHC-I expression in SCLC

We next aimed to elucidate the molecular pathways that regulate MHC-I expression in SCLC. We performed a comprehensive phospho-proteomic analysis and found that ERBB2 is strongly phosphorylated and associated with an increased phosphorylation of ERK in murine metastatic SCLC cell lines in comparison to SCLC cell lines derived from primary tumors (Fig. 3a, Supplementary Fig. 3a). Also at the protein level, human SCLC cell lines derived from meta-static sites showed higher ERBB2 expression compared to those derived from primary lung tumors (Fig. 3b). To investigate potential mechanisms of ERBB2 signaling in metastatic SCLC, we performed RNA sequencing on primary and metastatic murine SCLC, as well as analyzed publicly available scRNA-seq data from SCLC patients[10]. We assessed the expression of potential ERBB2 regulators in murine primary metastatic samples. No significant differential expression was observed in genes associated with the EGFR, WNT or NOTCH signaling pathways (Supplementary Fig. 2a). Consistent with our murine findings, analysis of human primary and metastatic SCLC samples revealed no upregulation of EGFR, WNT or NOTCH related gene sets in metastases (Supplementary Fig. 2b). Furthermore, whole-exome sequencing of matched primary and metastatic SCLC cells showed no mutations in *ERBB2*, *EGFR*, *WNT*, or *NOTCH* pathway genes (Supplementary Data 1 and 2). We further performed copy number variation analysis and found no amplification in *ERBB2* gene (Supplementary Fig. 2c). We further applied mass spectrometry to conduct global- and phospho-protein profiling in metastatic and primary SCLC (Fig. 3c, Supplementary Fig. 3b). We observed an overexpression of proteins associated with MAPK and AKT signaling as known downstream pathways of ERBB2 in metastatic SCLC, compared to primary SCLC (Fig. 3c). Moreover, proteins associated with immune-related genes, such as RIG-I or TBK1 as well as antigen presentation pathway proteins as TAP1 are enhanced in primary lung SCLC in comparison to SCLC liver metastases (Fig. 3c). We next analyzed phospho-proteins that are related to downstream signaling of ERBB2 and found increased phosphorylation of MAP2K2, RAF1, SHC1, GAB2, or EIF4B associated with MAP Kinase and AKT activity in metastatic SCLC (Supplementary Fig. 3b). These first results indi-cated ERBB2 signaling to be associated with downregulation of MHC-I via AKT and MAPK signaling. At the same time, we observed a loss of MAVS phosphorylation in SCLC metastasis, a key adaptor protein downstream of RIG-I signaling pathway[12] (Supplementary Fig. 3b, Supplementary Data 3 and 4). To validate these findings, we next fully knocked out *ERBB2* on SCLC cells (Fig. 3d, e). ERBB2 KO in SCLC cells did not affect SCLC proliferation (Supplementary Fig. 4c, d) but was accompanied by a significant increase of MHC-I expression (Fig. 3f). Of note, rescue of ERBB2 in ERBB2 KO SCLC again abrogated the increase of MHC-I, supporting the hypothesis that ERBB2 reg-ulates MHC-I expression (Fig. 3g, Supplementary Fig. 5h). To corro-borate these findings, we injected ERBB2 KO tumor cells orthotopically into immunocompetent mice. Strikingly, ERBB2 KO significantly prolonged survival compared to wild type SCLC cells and prevented metastatic spread into the liver (Supplementary Fig. 6a–c). In line with these findings, injected intravenously, the ERBB2 KO significantly reduced SCLC metastasis formation (Sup-plementary Fig. 6d–f). We next aimed to investigate whether the anti-tumor effect of ERBB2 inhibition is T cell dependent. Strikingly, CD4 and CD8 T cell depletion nearly completely abrogated the inhibitory effect of ERBB2 KO on SCLC tumor growth in immunocompetent mice (Supplementary Fig. 7a, b). In contrast, in the absence of an intact immune system in NSG (NOD scid gamma) mice, WT and

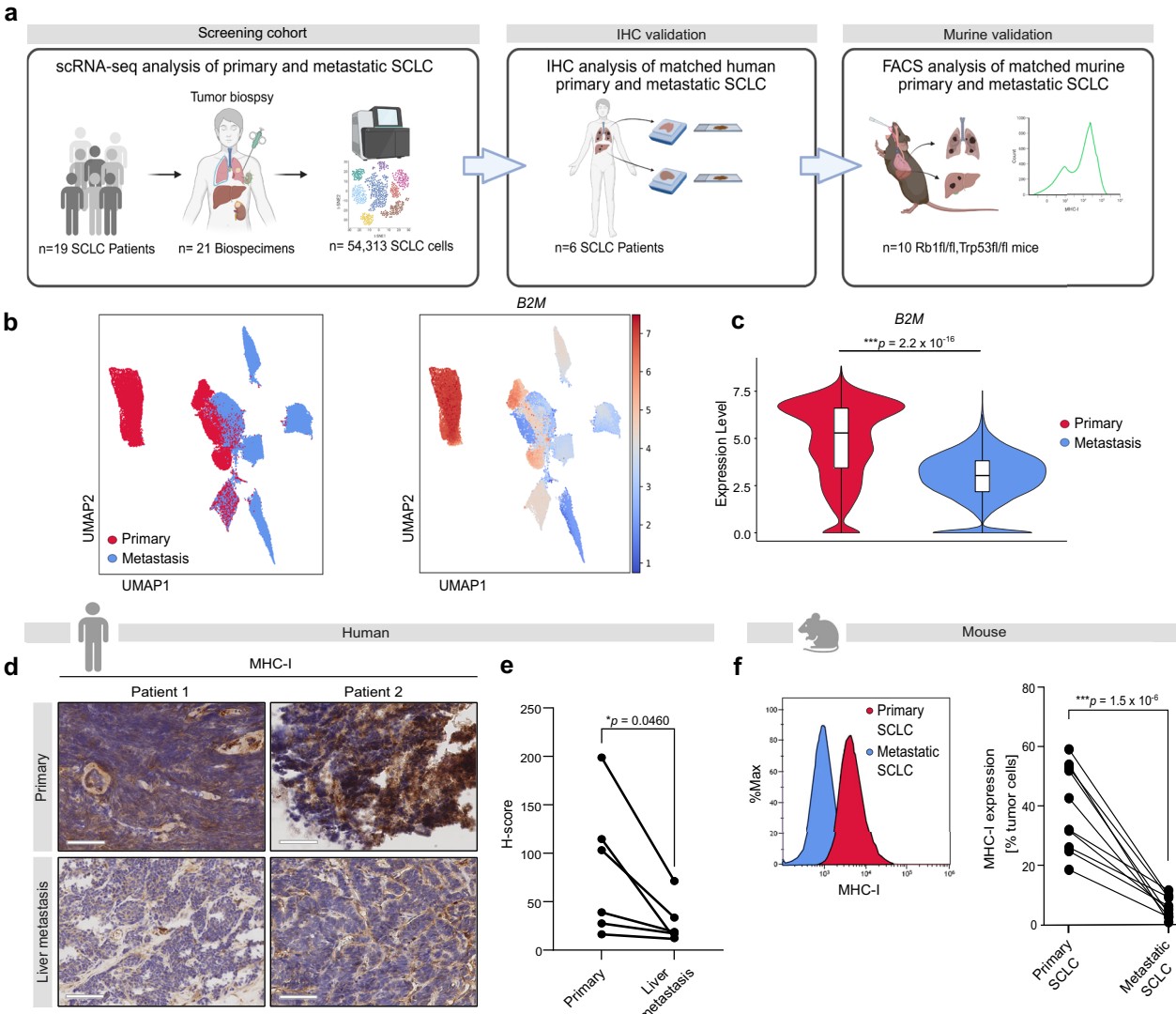

**Fig. 1 | SCLC liver metastases present significantly reduced MHC-I expression compared to primary lesions. a** Schematic of the initial analysis of publicly available scRNA-seq data from SCLC primary tumors and metastases[10], followed by validation through the analysis of matched SCLC samples from primary tumors and liver metastases in human and murine context. Created in BioRender. Meder, L. (2025) https://BioRender.com/z2ehzfh. **b** UMAP of *B2M* expression in publicly available scRNA-seq data from SCLC primary tumors and metastases (21 biospecimens, 19 patients)[10]. **c** Expression of *B2M* in primary (*n* = 9) and metastatic (*n* = 12) SCLC patient samples in scRNA-seq data[10] (two-sided Mann–Whitney test). The center line represents the median, box bounds represent Q1 and Q3 percentiles and whiskers extend to the minimum and maximum values. For primary: Min = 0, Q1 = 3.4412, Median = 5.2901, Q3 = 6.0248, Max = 8.8348, *n* = 26,043 cells. For

metastasis: Min = 0, Q1 = 2.1763, Median = 3.0322, Q3 = 3.8150, Max = 7.5308, *n* = 28,270 cells. **d** Representative images of matched patient samples (*n* = 6 patients) from primary and liver metastases. Images were taken at 20× magnification. Scale bars, 60 μm. **e** Expression of MHC-I on patient samples quantified by *H*-score analysis on IHC with Qupath software (two-sided, paired Student's *t*-test; *n* = 6 patients). Clinicopathologic characteristics are listed in the Supplementary Data 10. **f** Exemplary histograms and corresponding quantification of MHC-I surface expression determined by flow cytometry in matched samples from primary and liver metastasis in *Rb1/Trp53*-depleted autochthonous mouse model (two-sided, paired Student's *t*-test; *n* = 10 mice). *\*p* < 0.05, \*\*\**p* < 0.001. Source data are provided as a Source Data file. Icons created in BioRender. Meder, L. (2025) https://BioRender.com/z2ehzfh.

ERBB2 KO tumors showed similar growth behavior indicating that ERBB2 KO does not affect intrinsic tumor cell proliferation (Supplementary Fig. 4a, b). We next performed RNA sequencing and found that genes related to peptide transporters associated with antigen processing (*TAP1*) and an interferon gamma-mediated inflammatory response, such *H2-K1*, which is coding for an MHC-I subunit, were strongly enriched upon ERBB2 KO in SCLC cells (Fig. 3f, Supplementary Data 5). Upon ERBB2 KO in SCLC cells, we further found an induction of RIG-I (encoded by *DDX58*) that is known to elicit IFN stimulatory signaling (Fig. 3h)[12]. On protein level in ERBB2 KO, proteins involved in antigen processing (PSMB10, PSME1, PSME2), innate immune sensing (RIG-I, ISG15) and endoplasmic reticulum chaperone function (CALR, CANX) were upregulated, indicating a potential

restoration of immunogenic signaling pathways[12] (Fig. 3i, Supplementary Fig. 3i). Consistently, phosphoproteomic analysis revealed increased phosphorylation of TBK1 and CANX, accompanied by a reduction in phosphorylation of key downstream effectors of ERBB2 signaling, including MAP3K1, RAF1, SHC, and GAB1/2 (Supplementary Fig. 3c, d). In line with these findings, pharmacological inhibition of ERBB2 with mubritinib and neratinib led to upregulation of antigen processing and innate immune genes, including PSME1, PSMB8, IRF3, H2-K1 and RIG-I accompanied by downregulation of AKT and MAPK signaling pathways (Fig. 3j). Similarly, treatment with lapatinib resulted in increased expression of STING, PSMB8, TAP1, CALR, B2M, RIG-I, PSMB10, H2-K1 and H2-D1, further indicating enhanced antigen presentation and immune activation

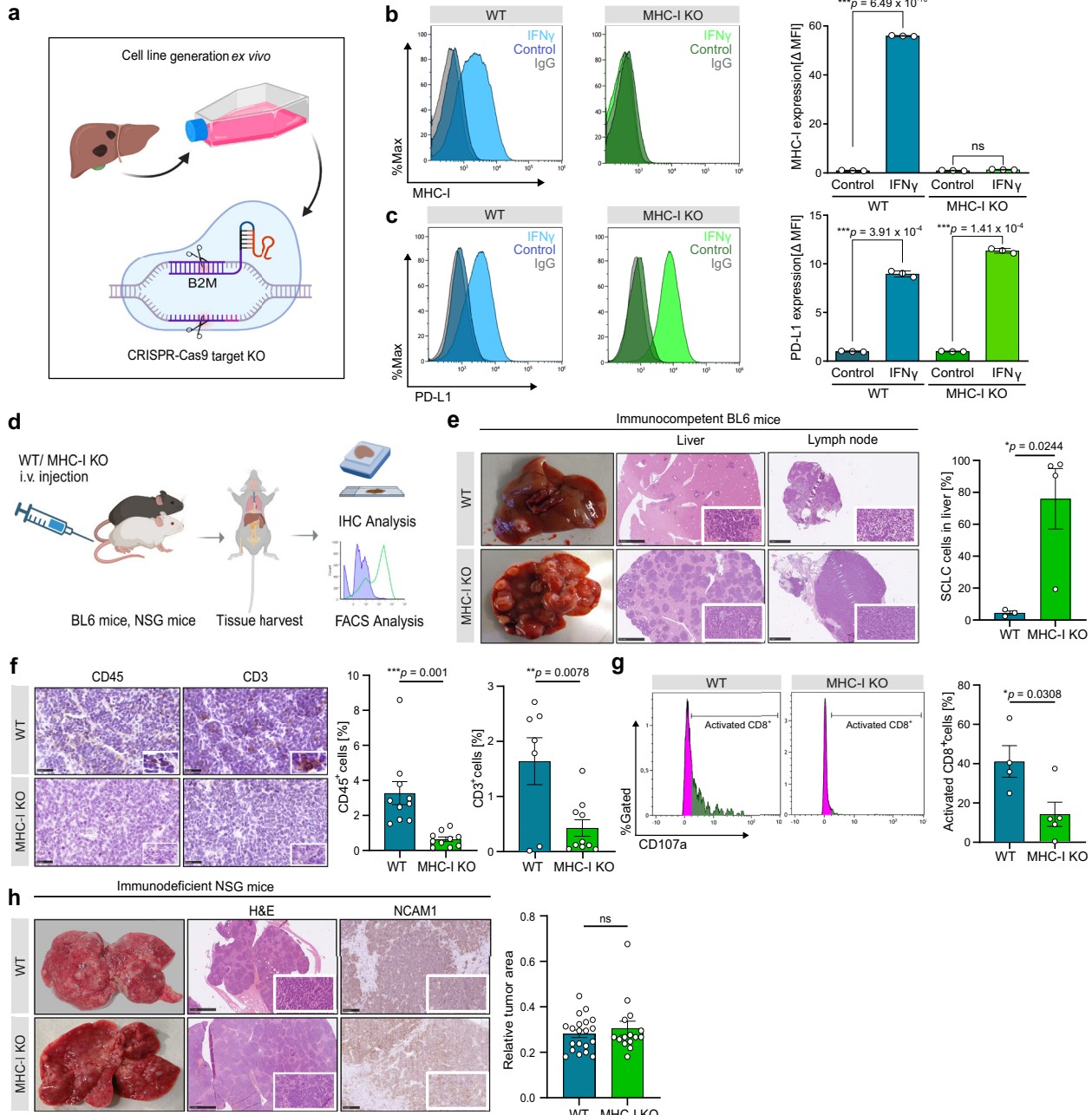

**Fig. 2 | MHC-I knockout promotes immune cell evasion and metastasis in SCLC mouse models. a** Schematic of B2M KO generation in murine SCLC primary cell line. Created in BioRender. Meder, L. (2025) https://BioRender.com/4u5k5v3. **b, c** Relative MHC-I or PD-L1 expression of WT and MHC-I KO cells analyzed by flow cytometry, determined by mean fluorescence intensity (MFI) normalized to IgG control. Histograms of one representative experiment are shown ($n = 3$ biological replicates). **d** Schematic of experimental setup showing intravenous injection of MHC-I KO and WT cells into immunocompetent C57BL/6/immunodeficient NSG mice, tissue harvest and subsequent IHC and FACS analyses. Created in BioRender. Meder, L. (2025) https://BioRender.com/w6kb2vb. **e** Representative images of H&E-stained livers and intestinal lymph node tissues from iv. WT and MHC-I KO injected immunocompetent mice. Scale bars, liver 2.5 mm, for lymph nodes 1 mm. Additional FACS based quantification of liver tumor cell infiltration ($n = 4$ mice per group). **f** Representative images of CD45 and CD3 IHC staining in liver tissue from WT and MHC-I KO-injected mice. Quantification was performed using QuPath analysis with 5 regions of interest (ROIs) analyzed per individual ($n = 2$ per group). **g** Representative histograms showing CD107a expression measured by flow cytometry and corresponding quantification (WT $n = 4$ mice, MHC-I KO $n = 5$ mice). **h** Representative images of H&E- and NCAM1-stained liver tissue of MHC-I KO and WT injected immunodeficient NSG mice (MHC-I KO $n = 3$, WT $n = 4$) and quantification of tumor cell infiltration in 5 representative ROIs by QuPath analysis. Scale bars, H&E 2.5 mm, NCAM1 100 μm. Statistical analysis was performed using a two-sided, unpaired Student's *t*-test. Data in this figure are shown as mean ± SEM. ns not significant, *$p < 0.05$, **$p < 0.01$, ***$p < 0.001$. Source data are provided as a Source Data file.

(Supplementary Fig. 3j–l). Phosphoproteomic analysis under ERBB2 inhibition revealed downregulation of AKT and MAPK signaling pathways, accompanied by enhanced innate immune sensing and interferon regulatory signaling, as indicated by increased phosphorylation of MAVS and IKBKE, respectively (Supplementary Fig. 3e–h). Strikingly, we could demonstrate that therapeutic inhibition of ERBB2 strongly induces MHC-I expression in different human and murine SCLC cell lines with and without IFNγ co-stimulation (Fig. 3k, l; Supplementary Fig. 5a–c) without affecting neuroendocrine differentiation (Supplementary Fig. 8). As

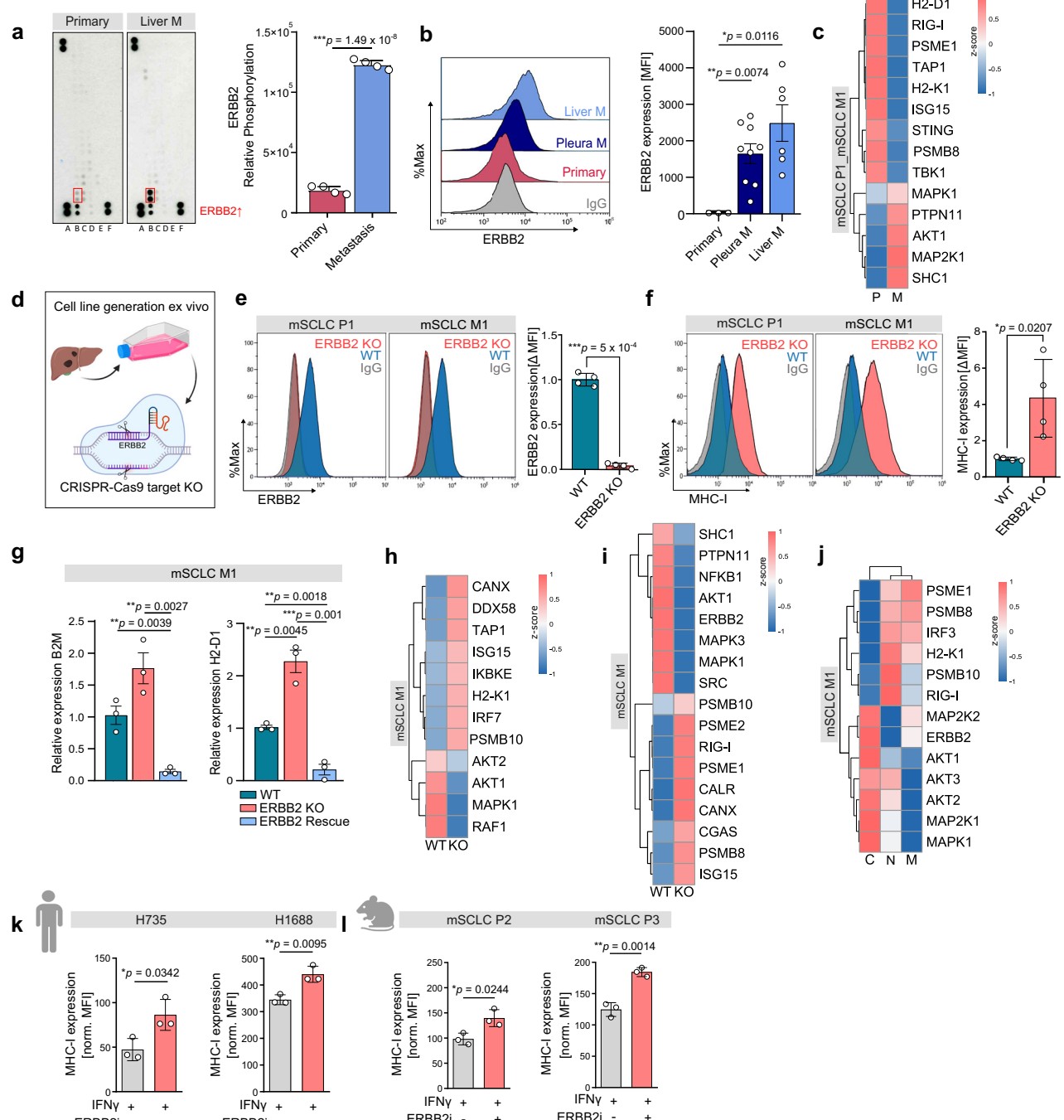

**Fig. 3 | ERBB2 regulates MHC-I antigen presentation pathway and inhibition of ERBB2 enhances MHC-I expression in SCLC. a** Relative phosphorylation of ERBB2 in primary and metastatic cells assessed by RTK-assay (n = 4 technical replicates). **b** Histogram showing ERBB2 expression determined by MFI in flow cytometry in representative human SCLC cell lines from primary tumor (*n* = 1), pleural effusion (*n* = 3), and liver metastases (*n* = 2) with corresponding quantification (*n* = 3 technical replicates). **c** Heat map of global protein profiling from primary and metastatic SCLC cells (each group *n* = 3 biological replicates). **d** Illustration of ERBB2 KO generation. Created in BioRender. Meder, L. (2025) https://BioRender.com/su67ybp. **e** Relative ERBB2 expression of WT and ERBB2 KO cells, determined by flow cytometry (*n* = 4 biological replicates). **f** Relative MHC-I expression of WT and ERBB2 KO cells, determined by flow cytometry (*n* = 4 biological replicates). **g** Relative *B2M*, *H2-D1* expression in WT, ERBB2 KO and ERBB2 rescue in murine metastatic SCLC cell line measured by qPCR (*n* = 3 biological replicates). **h** Heat

map of RNA sequencing in WT and ERBB2 KO cells (each group *n* = 3 biological replicates). **i** Whole proteome analysis of WT and ERBB2 KO in murine metastatic SCLC cell line (each group *n* = 3 biological replicates). **j** Whole proteome analysis of murine metastatic SCLC cell line untreated vs. treated with neratinib/mubritinib (1 μM) for 24 h (each group *n* = 3 biological replicates). **k** Relative MHC-I expression after treatment with IFNγ (40 ng/mL) or IFNγ + ERBB2 inhibitor mubritinib (1 μM), determined by flow cytometry (*n* = 3 biological replicates). **l** Relative MHC-I expression after treatment with IFNγ (40 ng/mL) or IFNγ + ERBB2 inhibitor mubritinib (1 μM), determined by flow cytometry (*n* = 3 biological replicates). For flow cytometry analyses, MFI was normalized to IgG control and representative histograms are shown. Statistical analysis was performed using a two-sided, unpaired Student's *t*-test. Data are represented as mean ± SEM. \**p* < 0.05, \*\**p* < 0.01, \*\*\**p* < 0.001. Source data are provided as a Source Data file. Icons created in BioRender. Meder, L. (2025) https://BioRender.com/z2ehzfh.

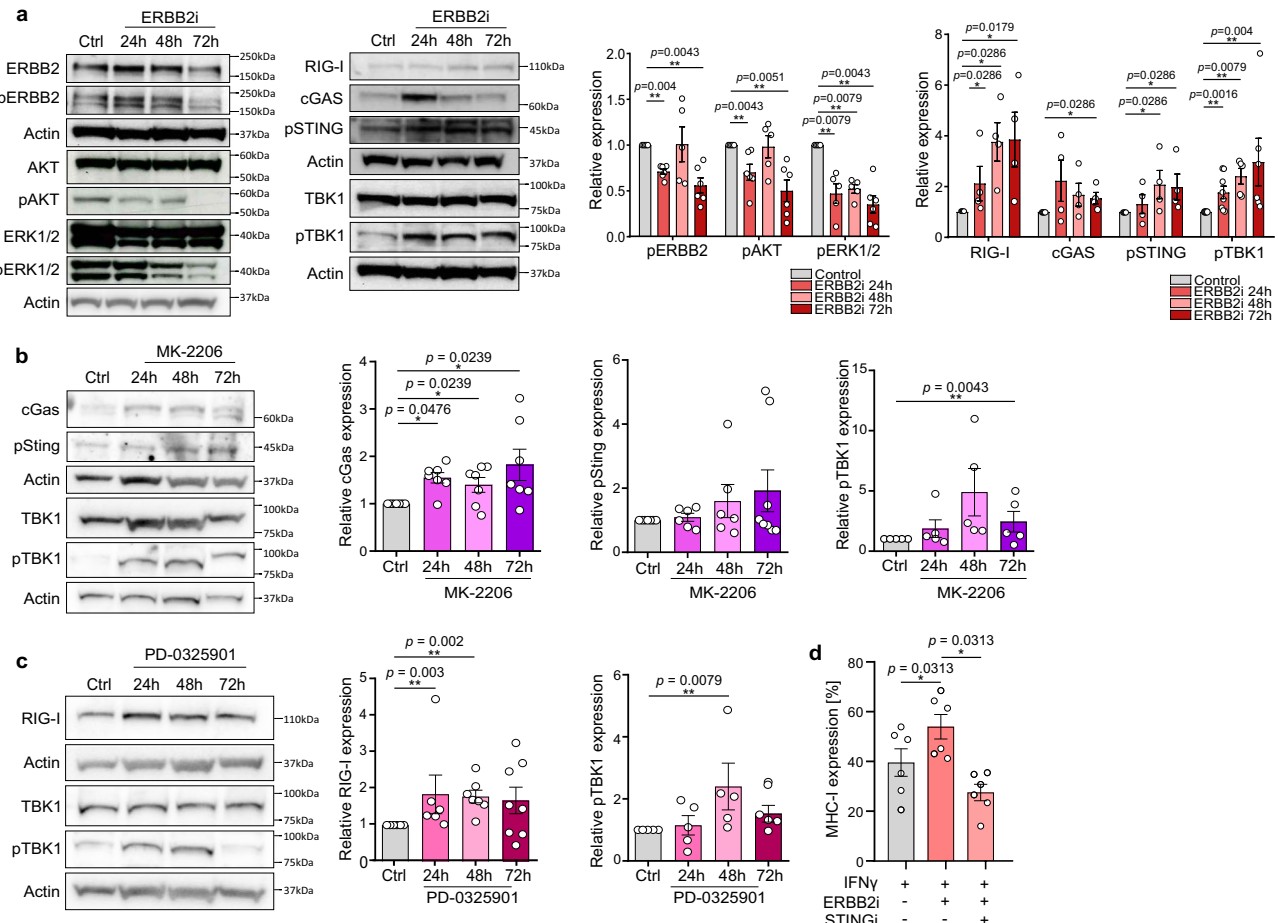

**Fig. 4 | ERBB2 regulates MHC-I via MAPK-AKT-TBK1 signaling. a** Western blot and corresponding quantification of relative protein quantity and phospho-protein status in mSCLC P1 treated with DMSO (1 µL, control) or ERBB2i (mubritinib, 1 µM) for 24–72 h (two-sided Mann–Whitney test; $n = 4$ and 5 biological replicates, respectively). **b** Western blot and corresponding quantification in mSCLC P1 treated with DMSO (1 µL, control) or MK-2206 (1 µM) for 24–72 h (two-sided Mann–Whitney test; $n = 6$). **c** Western blot and corresponding quantification in mSCLC P1 treated with of DMSO (1 µL, control) or PD-0325901 (1 µM) for 24–72 h (two-sided Mann–Whitney test; $n = 4$ and 5, respectively). **d** Percent of relative MHC-I expression after treatment with IFNγ only (40 ng/mL), IFNγ + ERBB2 inhibitor mubritinib (1 µM) or IFNγ + ERBB2 inhibitor mubritinib + STINGi (1 µM each) (two-sided, unpaired Student's t-test; $n = 6$ biological replicates per group). Western blots were quantified by densitometry and band intensities were normalized to β-Actin and expressed relative to the control group. Data are represented as mean ± SEM. *$p < 0.05$, **$p < 0.01$. Source data are provided as a Source Data file.

mubritinib has been shown to also target the electron transport chain[13], we validated our findings with the ERBB2 inhibitors neratinib and lapatinib and could demonstrate in analogy to mubritinib an increase in MHC-I expression (Supplementary Fig. 5a–c). Additionally, treatment of ERBB2 KO with mubritinib did not lead to increased MHC-I expression, supporting an ERBB2 dependent regulation of MHC-I (Supplementary Fig. 5f). In summary, these data strongly indicate that ERBB2 signaling induces loss of MHC-I in metastatic SCLC and that ERBB2 inhibition rescues MHC-I expression in SCLC.

### ERBB2 signaling mediates inflammatory programs in SCLC

We next investigated the effect of signaling pathways downstream of ERBB2 on inflammatory programs. In SCLC cells derived from primary SCLC tumors, treatment with an ERBB2 inhibitor resulted in repression of phosphorylation of AKT and ERK1/2, indicating an inhibitory effect on AKT and ERK1/2 signaling (Fig. 4a). We further investigated the effect of ERBB2 inhibition on viability of SCLC cells measured by Annexin V/PI staining (Supplementary Fig. 4e). Here we did not observe a decrease in cell viability upon ERBB2 inhibition. Innate immune sensors, such as RIG-I and cGAS have been described as mediators of immune cell-related death of cancer cells[14–16] and promoters of TBK1 pathways[17,18]. Upon drug-induced blockade of

ERBB2, we observed an increase in the phosphorylation of TBK1 in a time-dependent manner, which was associated with an increase in the relative expression of RIG-I and cGAS and the phosphorylation of STING (Fig. 4a, Supplementary Fig. 9a). This effect was not immediately apparent after short durations (10 min, 30 min, 2 h, 4 h; Supplementary Fig. 9e), but we observed a delayed induction at later time points (24 h, 48 h, 72 h; Fig. 4a). To further decipher the impact of downstream signaling of ERBB2 pathways on TBK1, we inhibited AKT or ERK1/2 with specific AKT and ERK inhibitors (Supplementary Fig. 9b–d). Inhibition of AKT resulted in a modest induction of cGAS (Fig. 4b, Supplementary Fig. 9b) and inhibition of ERK induced RIG-I expression (Fig. 4c, Supplementary 9b). These results indicate that ERBB2 inhibits TBK1 and cGAS primarily via AKT and RIG-I via MAP kinase signaling. Since ERBB2 blockade induces phosphorylation of STING, we investigated whether MHC-I expression upon ERBB2 blockade is mediated by STING. Pharmacological inhibition of STING prevented expression of MHC-I after ERBB2 blockade, particularly in a co-stimulatory setting with IFNβ. However, STING inhibition was able to reduce MHC-I expression on SCLC cells stimulated with IFNβ or IFNγ (Fig. 4d, Supplementary Fig. 5c–e). Also STING inhibition in ERBB2 KO could counteract MHC-I upregulation (Supplementary Fig. 5g). Thus, our data suggest that ERBB2 blockade induces MHC-I

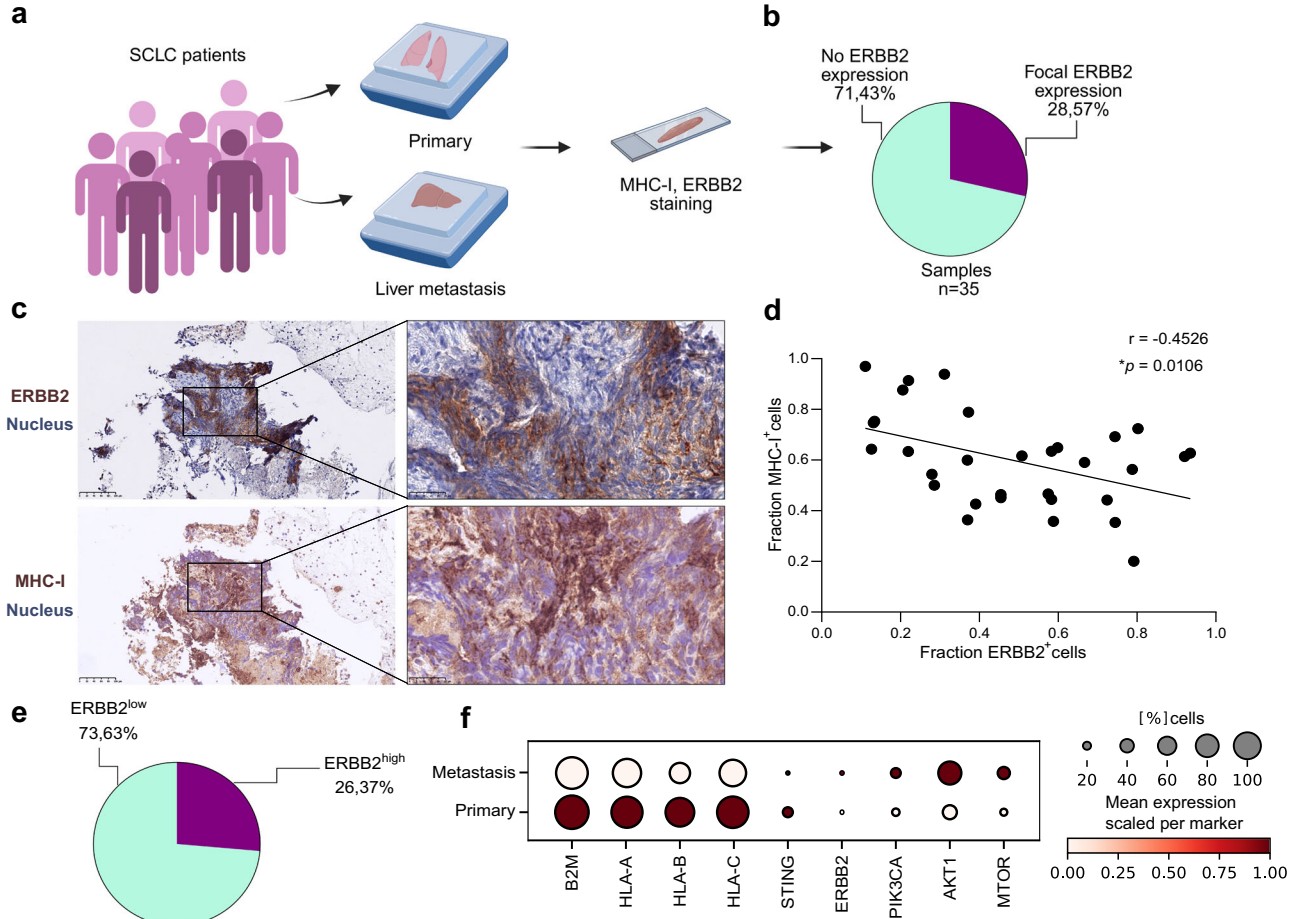

**Fig. 5 | ERBB2 expression in SCLC in patients. a** Schematic of FFPE samples of unmatched lung and liver samples from SCLC patients ($n = 35$) stained for ERBB2 and MHC-I in IHC. Created in BioRender. Meder, L. (2025) https://BioRender.com/x5oxh7c. **b** Proportional distribution of ERBB2-negative and ERBB2-focally positive patient samples. **c** ERBB2 and MHC-I IHC staining of one representative patient case. Images were taken at 20× magnification. Scale bars, 100 μm and 25 μm, respectively. **d** Correlation of ERBB2 and MHC-I expression based on QuPath-guided IHC analysis of representative ROIs from four patients. Clinicopathologic characteristics are listed in the Supplementary Data 10. A Pearson correlation analysis was performed, and a linear regression line was added to the graph. **e** Proportional distribution of ERBB2 low and ERBB2 high patient samples (low ERBB2 < 6%, high ERBB2 > 6%) in scRNA-seq data[10]. **f** Dot plot of scRNA-seq data[10] comparing primary tumors ($n = 9$) to metastatic sites, including liver, pleural effusion, adrenal gland and axillary lymph nodes ($n = 6$). $*p < 0.05$. Source data are provided as a Source Data file.

expression in a STING dependent manner. We next validated our findings in a cohort of 35 SCLC patients with unmatched lung and liver samples. We observed focal ERBB2 expression in 28.57% of patients with SCLC (Fig. 5a, b). IHC staining of ERBB2 and MHC-I revealed an inverse correlation between ERBB2 and MHC-I expression (Fig. 5c, d). We additionally analyzed transcriptomic data from human SCLC cell lines[19] and could again confirm an inverse correlation between ERBB2 and B2M (Supplementary Fig. 10a). In line with our findings, the analysis of publicly available scRNA-seq datasets[10] revealed a comparable frequency of patients with high ERBB2 in expression samples in 26.37% (Fig. 5e). Additionally, we analyzed the IMpower133 data set and observed a tendency that high ERBB2 expression is associated with a shorter overall survival in the immunochemotherapy patient cohort (Supplementary Fig. 10b). As we identified ERBB2 as a relevant pathway that mediates the expression of MHC-I and thereby antigen presentation in SCLC, we further investigated the impact of ERBB2 expression in SCLC patients with high tumor mutational burden (TMB). TMB has been shown to positively correlate with the number of neoantigens whereas its presentation is mediated by tumor intrinsic pathways[20] such as ERBB2 in our study. Corroborating our findings, we observed in the

subgroup analysis of patients with high TMB, that high ERBB2 expression significantly correlates with worse outcome after combined immune checkpoint inhibition with chemotherapy (Supplementary Fig. 10c). These data indicate that ERBB2 signaling influences the presentation of neoantigens in SCLC patients. Comparing primary tumors with metastatic lesions in scRNA-seq data further demonstrated a loss of *B2M*, *HLA* genes *A*, *B*, *C* and *STING* expression alongside upregulation of *ERBB2* and downstream AKT signaling pathway in metastasis (Fig. 5f).

### Inhibition of ERBB2 enhances T cell-mediated immune response and prevents metastatic immune cell evasion

Motivated by the finding that ERBB2 blockade induces MHC-I expression, we applied a fully autochthonous SCLC mouse model to investigate the impact of drug-induced ERBB2 inhibition on SCLC immune cell evasion and metastasis (Fig. 6a). Strikingly, treatment with an ERBB2 inhibitor drastically reduced the formation of liver metastasis (Fig. 6b), whereas ERBB2 knock-out in vitro did not affect SCLC tumor cell proliferation (Supplementary Fig. 4c, d). In parallel, we harvested tumors from lungs, livers and lymph nodes and quantified the ratio of immune cells by FACS analysis. In each organ,

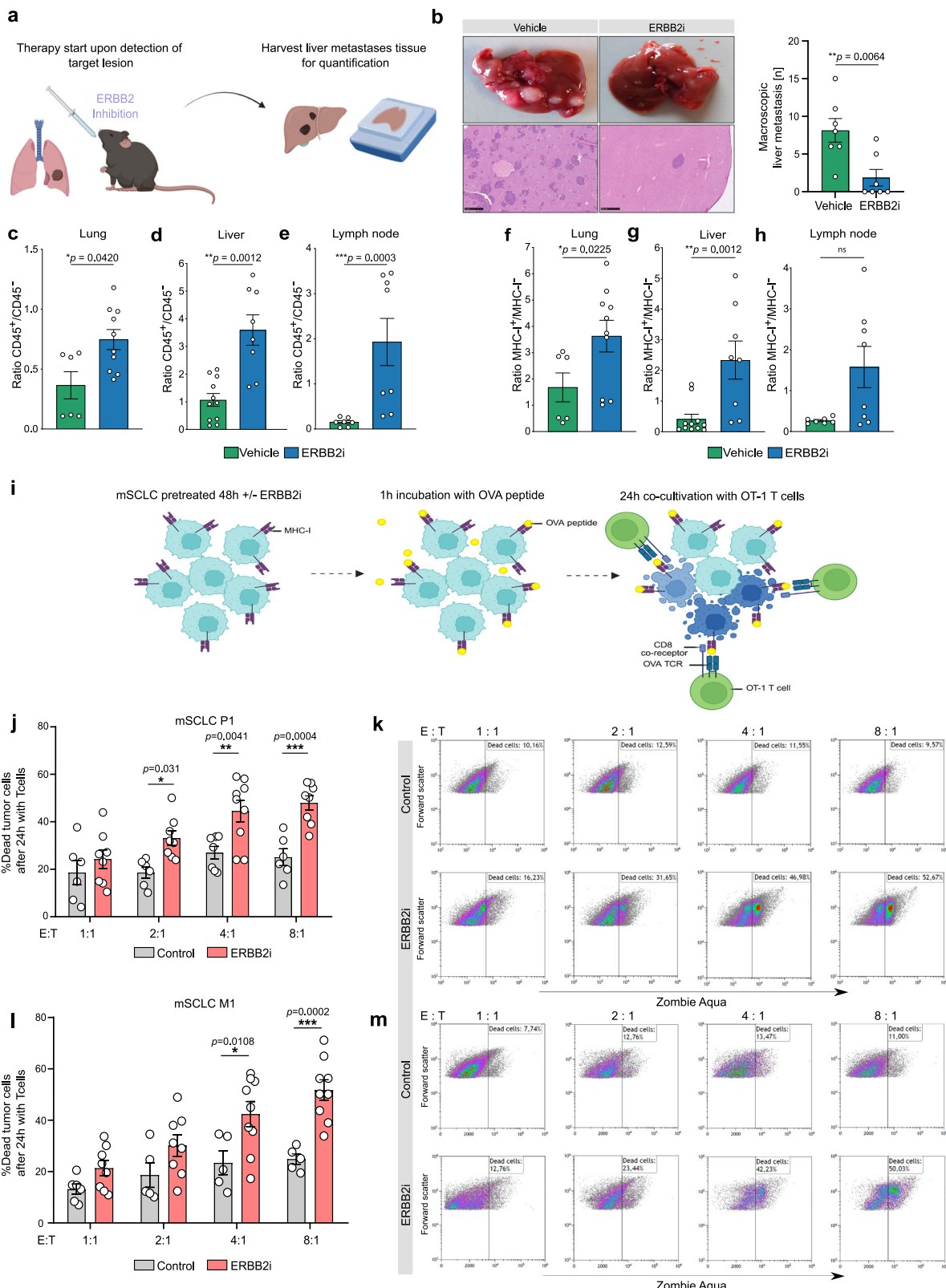

the quantities of CD45+ immune cells were significantly increased in SCLC mice treated with the ERBB2 inhibitor, compared to vehicle-treated SCLC mice (Fig. 6c–e). We further assessed the expression of MHC-I and found an increased ratio of MHC-I positive cells in SCLC mice treated with an ERBB2 inhibitor (Fig. 6f–h). Taken together, these data strongly strengthen the assumption that inhibition of ERBB2 prevents SCLC immune cell evasion and thereby the

formation of liver metastasis. We next examined whether ERBB2 blockade induced MHC-I expression triggers T cell-mediated SCLC cell killing. To investigate the impact of low MHC-I expression in SCLC cells, we pulsed SCLC cells with ovalbumin (OVA) peptide (SIINFEKL) that binds to cell surface MHC-I. We washed the cells to remove unbound peptide and then performed a co-culture with TCR-transgenic OT-I T cells that specifically recognize H-2Kb bound

**Fig. 6 | Inhibition of ERBB2 enhances T cell- mediated immune response and prevents metastatic immune cell evasion. a** Schematic of the experimental setup, including mouse treatment with ERBB2 inhibitor mubritinib and sample preparation. Created in BioRender. Meder, L. (2025) https://BioRender.com/eyhkfjh. **b** Representative macroscopic and microscopic images of liver tissue from vehicle and ERBB2i-treated mice with macroscopic quantification of liver lesions (two-sided Mann–Whitney test; $n = 7$ mice per group). **c–e** Ratio of CD45+/CD45− cells determined by flow cytometry from lung, liver and lymph node samples of vehicle or ERBB2i treated mice (two-sided Mann–Whitney test, lung, vehicle $n = 6$, ERBB2i $n = 10$; liver, vehicle $n = 11$, ERBB2i $n = 8$; lymph node, vehicle $n = 7$, ERBB2i $n = 8$). **f–h** Ratio of MHC-I+/MHC-I− tumor cells determined by flow cytometry from lung, liver, and lymph node samples of vehicle or ERBB2i (mubritinib) treated mice (two-sided Mann–Whitney test; lung, vehicle $n = 6$, ERBB2i $n = 10$; liver, vehicle $n = 11$, ERBB2i $n = 8$; lymph node, vehicle $n = 7$, ERBB2i $n = 8$). **i** Schematic of the tumor cell/ T cell co-culture assay setup. Created in BioRender. Meder, L. (2025) https://BioRender.com/3wkefcf. **j** Percent of dead target primary tumor cells following 24 h incubation with OT-I T cells at indicated effector/T cell (E:T) ratio with or without ERBB2i (mubritinib) (two-way ANOVA; 1:1 control $n = 6$, ERBB2i $n = 8$; 2:1 control $n = 6$, ERBB2i $n = 8$; 4:1 control $n = 7$, ERBB2i $n = 9$; 8:1 control $n = 6$, ERBB2i $n = 8$ biological replicates). **k** Representative dot plots showing gated cancer cells with dead cells labeled with Aqua Zombie. **l** Percentage of dead metastatic tumor cells after 24 h co-culture with OT-I T cells at indicated effector/tumor cell (E:T) ratios (two-way ANOVA; 1:1 control $n = 6$, ERBB2i $n = 8$; 2:1 control $n = 5$, ERBB2i $n = 8$; 4:1 control $n = 5$, ERBB2i $n = 9$; 8:1 control $n = 5$, ERBB2i $n = 9$ biological replicates). **m** Representative dot plots showing gated cancer cells with dead cells labeled with Aqua Zombie. Data are presented as mean ± SEM. *$p < 0.05$, **$p < 0.01$, ***$p < 0.001$. Source data are provided as a Source Data file.

to OVA peptide (Fig. 6i). We found that SCLC cells were rather unresponsive to antigen-specific T cell killing, however, pre-treatment with an ERBB2 inhibitor restored effective T cell killing of SCLC by induction of MHC-I (Fig. 6j–m). Taken together, these data show that an ERBB2 inhibition leads to increased T cell-mediated tumor cell killing.

## Combined blockade of ERBB2 and PD-1 enhances T cell recruitment and induces antigen-specific T cell clonality in vivo

Encouraged by the finding that ERBB2 blockade enhances MHC-I induced T cell-mediated killing of SCLC cells, we sought to investigate the impact of ERBB2 and anti-PD-1 on the tumor immune cell compartment applying scRNA-seq in an autochthonous mouse model of SCLC. Mice were randomized in groups treated with vehicle, anti-PD-1 alone, ERBB2i alone or the combination of anti-PD-1 antibody and an ERBB2 inhibitor and tumors were harvested for scRNA-seq analyses (Fig. 7a, b). We observed a clear increase of immune cell infiltration, particularly of T cells after ERBB2 inhibition (Fig. 7c, d, k). Moreover, a combination of anti-PD-1 and ERBB2 targeted treatment again enhanced immune cell recruitment associated with a decline of epithelial cells (Fig. 7e, i). In contrast, the other treatment groups exhibited a higher proportion of epithelial cells and with reduced immune cell infiltration (Fig.7f, g, h). We next quantified markers in epithelial and immune cells for each therapy group. In line with the enhanced T cell recruitment, we could detect an increase of the expression of known MHC-I-related genes as *H2-K1* and *B2M*, as well as peptide transporters associated with antigen processing (*TAP1* and *TAP2*) after combined inhibition of ERBB2 and PD-1 (Fig. 7j). These findings were associated with a decline of epithelial cells indicating that this treatment triggers a strong stimulation of MHC-I expression and a drastic decrease of EPCAM positive SCLC cells (Fig. 7 e, i). In line with these observations, we found an increase in the expression of T cell activation markers, such as *CD8a* and reduced expression of T cell exhaustion markers such as *TIGIT* or *LAG3* (Fig. 7k). These data underscore that a combination of ERBB2 and anti-PD-1 targeted treatment strongly induces MHC-I expression and finally T cell recruitment and activation. The here portrayed T cell infiltration determined by scRNA-seq (Fig. 7) likely provides a more accurate representation of the immune response in the tumor microenvironment than flow cytometry analysis (Supplementary Fig. 11), as it captures cell-type specific changes and was here conducted in response to treatment in the animals. We next sought to unravel whether combined ERBB2 and PD-1 blockade mediates a clonal expansion of effective T cells and performed TCR sequencing. We observed that SCLC mice treated with an ERBB2 and anti-PD-1 inhibitor displayed a higher number of circulating effective T cells, in comparison to vehicle-treated SCLC mice and healthy control mice (Fig. 8a). Most strikingly, ERBB2 and PD-1 inhibition resulted in the formation of expanded T cell clones among cytotoxic T cells (CTL_Clonotype c.1 = 16.03%, CTL_Clonotype c.2 = 8.85%;

Fig. 8b, Supplementary Data 6–9). This was strengthened by the increased clonality index measured by inverted Shannon entropy (Fig. 8c). We further analyzed the TCR repertoire of anti-ERBB2 and anti-PD-1 versus vehicle-treated SCLC mice using the Morisita's overlap index (MOI), which accounts for MOI = 0.016. The MOI was used to quantify the similarity of TCR clonotypes based on CDR3 sequences and their frequency. An MOI value of 0.016 suggests a low overlap between the TCR repertoires of the ERBB2/PD-1 and the vehicle-treated group, indicating distinct clonal expansions following treatment. We found that in SCLC mice treated with an ERBB2 and anti-PD-1 antibody the most frequent cytotoxic clones showed an increased similarity in the sequence of CDr3 β-chain calculated with quality method and global alignment using the stringDist()-function of the Biostrings package (Fig. 8d). Moreover, ERBB2i + anti-PD-1 treated SCLC mice additionally revealed an expansion of *TRBV16* & *TRBJ2-7* and *TRBV14* & *TRBJ2-3* gene combinations in the TCR β-chain across different clonotypes (Fig. 8e, f). Of note, these dominant clones in SCLC mice during ERBB2 and anti-PD-1 treatment showed a significant increase in the expression of cytotoxic features as *GZMA* (Fig. 8g). Hereby, the most expanded cytotoxic clone in the vehicle group presented with a terminally exhausted T cell phenotype indicated by increased expression of *TIGIT* and *TOX* (Fig. 8h, i). These data indicate a polyclonal antigen-specific expansion of specific cytotoxic T cell clones upon combined ERBB2 and anti-PD-1 treatment in SCLC mice.

## ERBB2 blockade overcomes resistance against anti-PD-1 treatment and displays synergistic treatment effects in autochthonous Rb1/Trp53-depleted SCLC mice

We first treated autochthonous conditionally *Rb1/Trp53*-deleted SCLC mice with an anti-PD-1 antibody and observed only very short treatment response, followed by a rapid progressive disease, which is in line with the observation in patients with SCLC under anti-PD-1 monotherapy. Interestingly, in accordance with our previous data[21], we observed a significant reduction of MHC-I-positive SCLC tumor cells in anti-PD-1-refractory SCLC mice, indicating that loss of MHC-I mediates resistance against anti-PD-1 treatment (Fig. 9f). We thus speculated that the addition of an ERBB2 inhibitor might overcome anti-PD-1 resistance by restoring MHC-I expression in *Rb1/Trp53*-depleted SCLC tumors. To decipher potential synergistic treatment effects by combining ERBB2 and anti-PD-1 blockade, we performed a preclinical study in an autochthonous mouse model of SCLC, in which tumors are induced upon Cre-mediated biallelic deletion of *Rb1* and *Trp53*. We recorded the clinicopathologic parameters of SCLC bearing mice listed according to the applied therapy regimens. Mice were randomized and systemically treated with vehicle, an anti-PD-1 antibody, the ERBB2 inhibitor mubritinib, or with the combination of mubritinib and an anti-PD-1 antibody. ERBB2 blockade or anti-PD-1 blockade alone in SCLC-bearing mice did not improve PFS and OS in comparison with vehicle-treated mice (Fig. 9a–c). Very

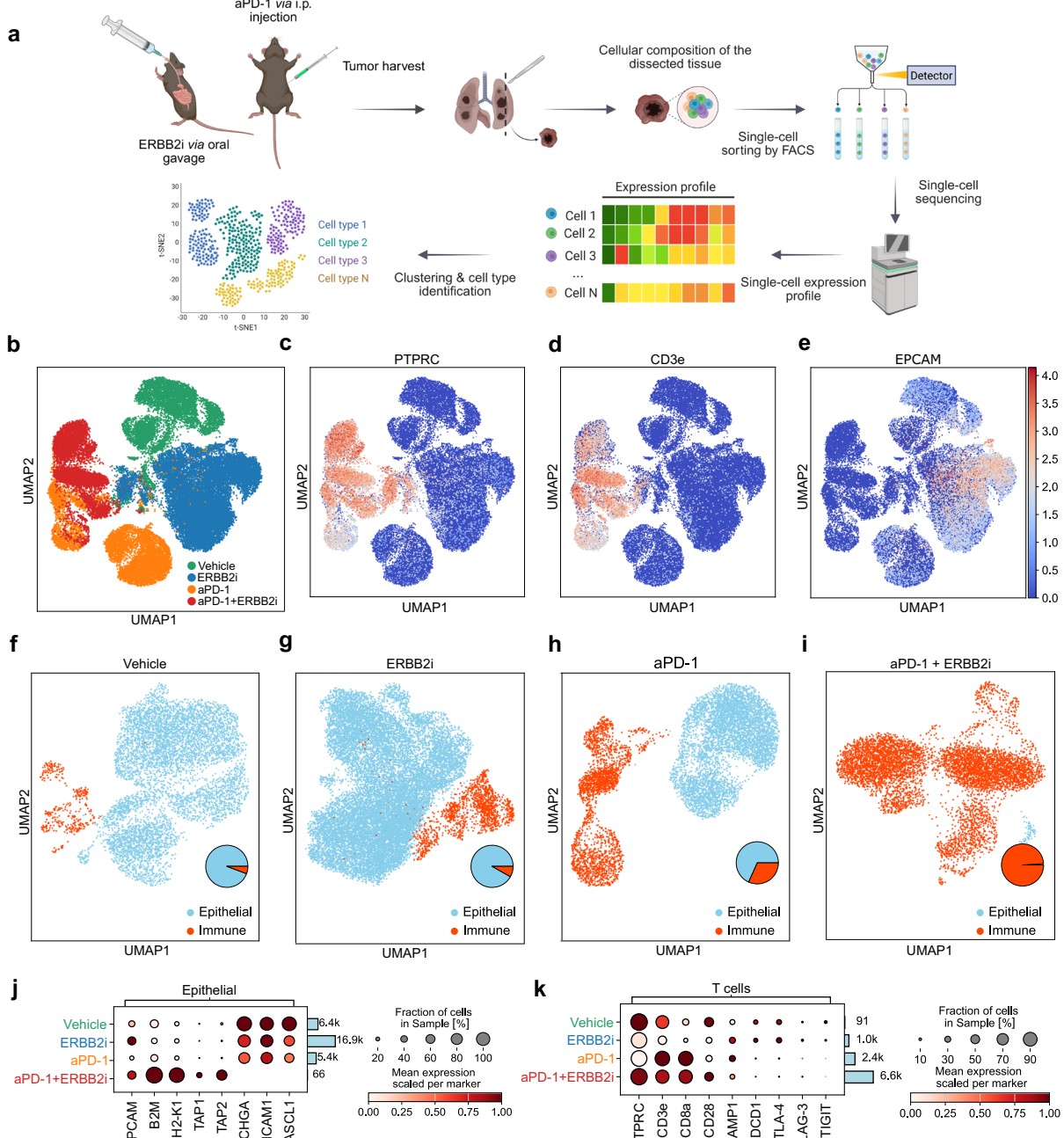

**Fig. 7 | Dual ERBB2 and PD-1 inhibition increases MHC-I-mediated antigen presentation and T cell recruitment. a** Schematic of the experimental setup showing treatment of autochthonous SCLC tumor-bearing mice, once lesions are detected by CT. Mice received either vehicle, anti-PD-1, ERBB2i (mubritinib), or a combination of anti-PD-1 and ERBB2i (mubritinib). After treatment, tumors were harvested and processed for scRNA-seq analysis. Created in BioRender. Meder, L. (2025) https://BioRender.com/bxcwg6c. **b** Treatment group annotation of cell clusters visualized in UMAP from scRNA-seq samples of four treatment conditions: vehicle (green), anti-PD-1 (orange), ERBB2i (blue), and combination anti-PD-1 and ERBB2i (red). **c–e** UMAP plots indicating expression of *PTPRC* (**c**), *CD3e* (**d**), *EPCAM* (**e**) in treatment groups (*n* = 4 mice). **f–i** Cell type identification of immune (orange) and epithelial tumor cells in vehicle (**f**), ERBB2i (**g**), anti-PD-1 (**h**) and anti-PD-1 + ERBB2i treated mice (**i**). The ratio of each cell fraction is represented in the pie chart in inset. **j** Dot plot of the mean expression of antigen-presentation-associated genes and neuroendocrine markers in the epithelial cells for each therapy group. **k** Dot plot of the mean expression per marker in the T cells for each therapy group. Source data are provided as a Source Data file.

strikingly, combined blockade of ERBB2i and anti-PD-1 led to a substantial improvement in median PFS and OS (Fig. 9b, c). Treatment with combined ERBB2 and anti-PD-1 blockade resulted in deep remissions with very good partial responses and one complete response (Fig. 9d). Importantly, we confirmed these results using an alternative ERBB2 inhibitor lapatinib in combination with anti-PD-1, which also significantly improved OS compared to anti-PD-1 mono-therapy (Supplementary Fig. 5i).

We further assessed the occurrence of liver metastasis and found that combined anti-ERBB2 + anti-PD-1 nearly completely abrogated the formation of liver metastasis highlighting the impact of ERBB2 and anti-PD-1 blockade to prevent SCLC immune cell evasion (Fig. 9e). We finally investigated the impact of ERBB2 and anti-PD-1 blockade on activated T cells and the expression of MHC-I on SCLC cells in vivo (Supplementary Fig. 11c, d; Fig. 9f). ERBB2 blockade alone resulted in a significant increase of activated CD4+ and CD8+ T cells in the tumor

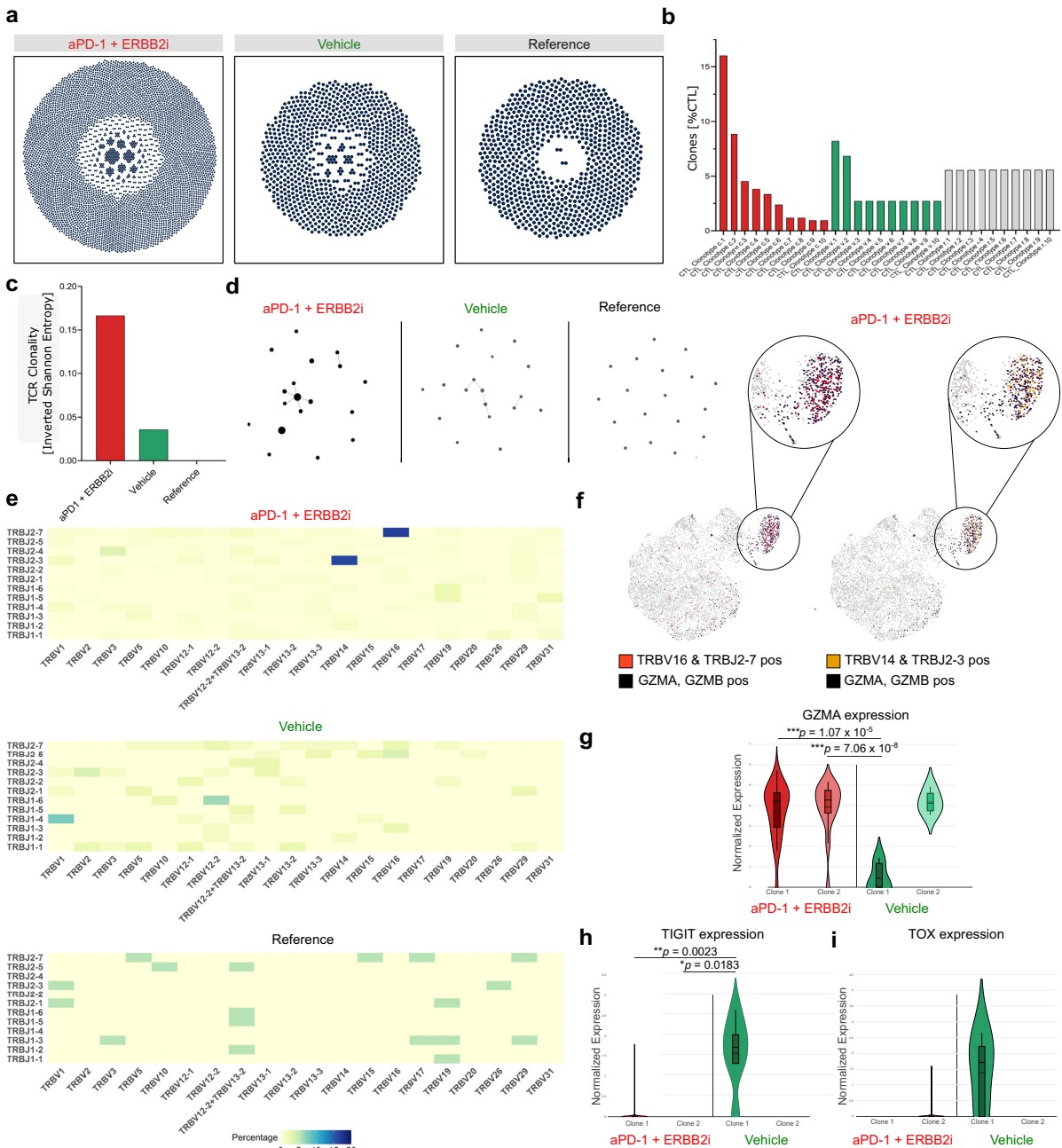

**Fig. 8 | Combined ERBB2 and anti-PD-1 blockade induces expansion of dominant T cell clones.** Peripheral T cells of SCLC-bearing mice in treatment and healthy control were analyzed. **a** Clonotype distribution of all T cells in anti-PD-1 + ERBB2i-treated, vehicle-treated and healthy control mice ($n = 1$ per group). **b** Clonotype frequency of the top 10 clones in cytotoxic lymphocytes. The respective group is indicated by color code. **c** TCR clonality measured by inverted Shannon entropy. **d** Cluster analysis of TCR β-chain, calculated with quality method and global alignment using the Biostrings R package. **e** V- and J-gene usage in TCR β-chain of different treated mice. **f** UMAP of all T cells with cytotoxic (blue), *TRBV14 & TRBVJ2*-3 positive (red) and *TRBV16 & TRBVJ2-7* positive (orange) T cells in ERBB2i + anti-PD-1 treated mice. **g**−**i** Comparison of the two most frequent cytotoxic clones in anti-PD-1 + ERBB2i (Clone 1 $n = 67$, Clone 2 $n = 37$) and vehicle (Clone 1 $n = 6$, Clone 2 $n = 4$) concerning *GZMA, TIGIT*, and *TOX* expression. The center lines represent the median or mean, box

bounds represent Q1 and Q3 percentiles and whiskers extend to the minimum and maximum values. For Gzma: anti-PD-1 + ERBB2i Clone 1 Min = 0, Q1 = 2.931559, Mean = 3.719872, Median = 4.206237, Q3 = 4.630662, Max = 5.726883, Clone 2 Min = 0, Q1 = 3.631347, Mean = 3.920051, Median = 4.268613, Q3 = 4.739704, Max = 5.163969; Vehicle Clone 1 Min = 0, Mean = 0.4342388, Q3 = 1.163976, Max = 1.441456, Clone 2 Min = 3.544061, Mean = 4.172555, Median = 4.112799, Q3 = 4.599101, Max = 4.920561. For Tigit: anti-PD-1 + ERBB2i Clone 1 Min = 0, Mean = 0.02606225, Median = 0, Max = 1.746171; Vehicle Clone 1 Min = 0, Q1 = 1.322435, Mean = 1.572579, Median = 1.719329, Q3 = 2.029191, Max = 2.645192. For Tox: aPD-1 + ERBB2i Clone 2 Mean = 0.042270, Median = 0, Max = 1.563993. Vehicle Clone 1 Min = 0, Mean = 1.720013, Median = 1.720013, Q3 = 2.225449, Max = 2.654358. Statistical analysis was done using the two-sided Mann–Whitney test. *$p < 0.05$, **$p < 0.01$, ***$p < 0.001$. Source data are provided as a Source Data file.

(Supplementary Fig. 11c, d). Moreover, we observed a trend towards upregulation of MHC-I and decreased ERBB2 expression on SCLC cells (Fig. 9f; Supplementary Fig. 11b). Also, an increase of MHC-II could be observed under ERBB2 inhibition (Supplementary Fig. 11a). In contrast, SCLC mice treated with anti-PD-1 antibody revealed a decreased MHC-I

expression on the tumor cells. Accordingly, analysis of scRNA-Seq data[10] revealed a downregulation of *B2M* in SCLC patients treated with anti-PD1 immunotherapy in combination with chemotherapy whereas *B2M* was maintained in patients treated with chemotherapy alone (Fig. 9g, h). Together, these data demonstrate that ERBB2 blockade

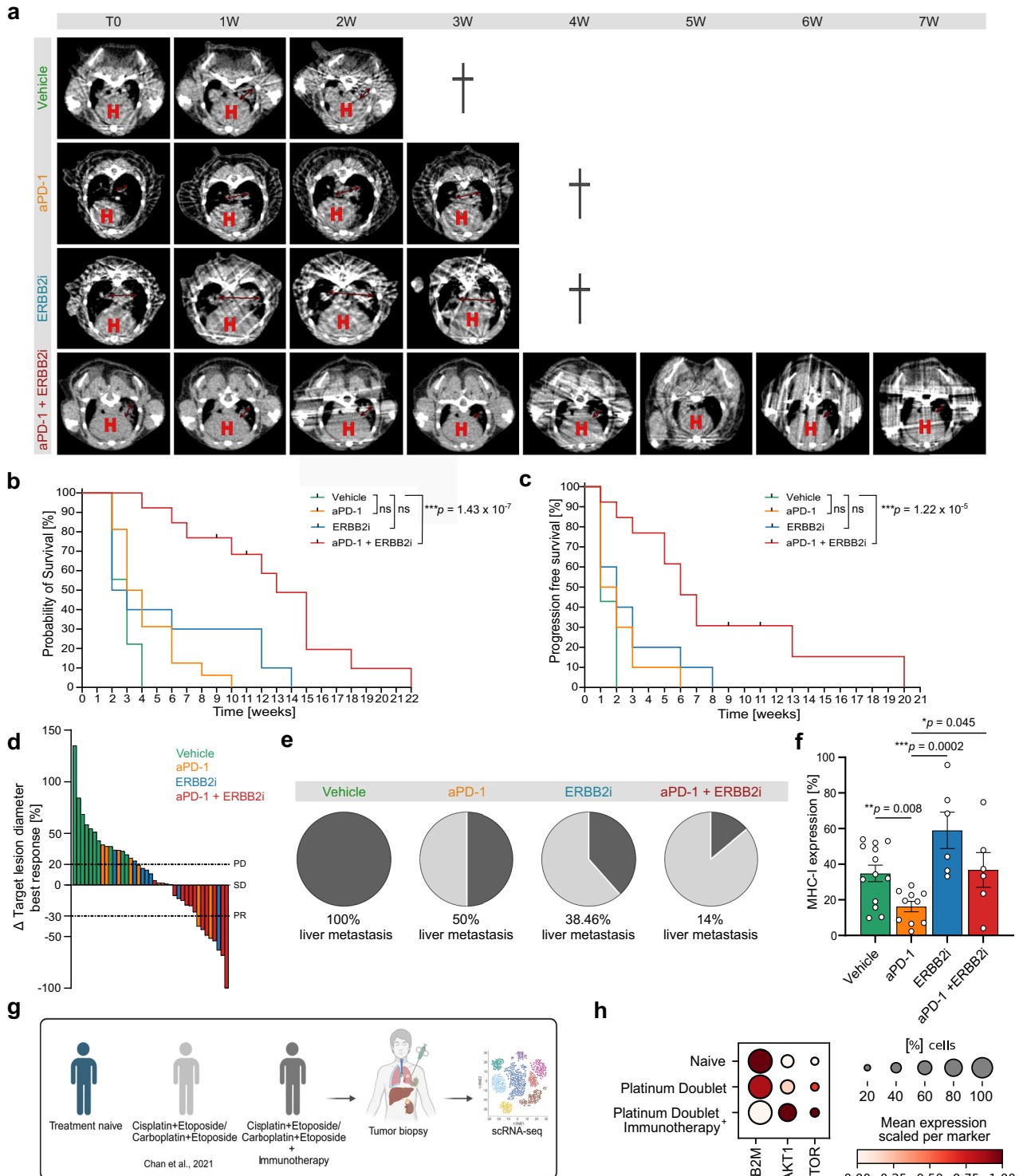

**Fig. 9 | ERBB2 inhibition combined with anti-PD-1 display synergistic treatment effects in SCLC mice. a** Serial μCT measurements of one representative mouse per therapy group. Target lesion diameter is marked in red. H, heart; cross (hand-drawn), dead. **b** OS was determined in the four therapy groups (log-rank (Mantel−Cox) test; vehicle $n = 18$; anti-PD-1 $n = 16$; ERBB2i (mubritinib) $n = 10$; anti-PD-1 + ERBB2i (mubritinib) $n = 13$). **c** PFS determined in the four therapy groups (log-rank (Mantel−Cox) test; vehicle $n = 14$; anti-PD-1 $n = 10$; ERBB2i (mubritinib) $n = 10$; anti-PD-1 + ERBB2i (mubritinib) $n = 13$). **d** Change in target lesion diameter calculated from all therapy groups after some weeks of treatment. PD, SD, and PR according to described mouse-adapted RECIST v1.1 criteria. **e** Pie chart representing the quantification of liver metastases after vehicle, anti-PD-1, ERBB2i or anti-PD-1 + ERBB2i treatment. **f** MHC-I expression determined by FACS analysis from lung of mice of the different treatment groups (two-sided Mann−Whitney test, vehicle $n = 13$, anti-PD-1 $n = 10$, ERBB2i $n = 6$, anti-PD-1 + ERBB2i $n = 6$; error bars, mean ± SEM). **g** Schematic representation of the analysis of different therapy groups in publicly available scRNA-seq data from SCLC patient tumors[10]. Created in BioRender. Meder, L. (2025) https://BioRender.com/vkn5pfz. **h** Dot plot of markers in different therapy groups in publicly available scRNA data set. *$p < 0.05$, **$p < 0.01$, ***$p < 0.001$. Source data are provided as a Source Data file.

increases MHC-I expression on SCLC cells in vivo and thereby prevents SCLC immune evasion and overcomes resistance against anti-PD-1 therapy in autochthonous *Rb1/Trp53*-deleted SCLC mice.

## Discussion

Here, we provide a mechanistic insight for the clinical observation of resistance against immune checkpoint inhibition in SCLC. We show that ERBB2-mediated repression of MHC-I expression induces immune evasion in SCLC mouse models, which is reflected by the loss of MHC-I expression in metastatic SCLC patients. Moreover, genetic or pharmacologic blockade of the ERBB2 signaling axis reinforces MHC-I expression and prevents immune evasion and metastasis formation in autochthonous murine SCLC. Finally, we demonstrate that the ERBB2 signaling axis regulates MHC-I expression on SCLC cells and is critical in maintaining immune evasion in SCLC. Most strikingly, combining ERBB2 with anti-PD-1 targeted treatment displays synergistic treatment efficacy with deep therapeutic responses in preclinical SCLC models.

The discovery of immune checkpoint molecules, such as PD-1, that mediate T cell inactivation and inhibit T cell function led to the development of immune checkpoint inhibitors reinforcing T cell-mediated killing of cancer cells[22,23]. In SCLC, biomarkers of an improved outcome under immunotherapy in the CASPIAN phase 3 trial (NCT03043872) included gene signatures related to the antigen-presenting and processing machinery and MHC-I expression in patients treated with anti-PD-L1 and anti-CTLA4[24]. However, the majority of patients with SCLC are resistant against checkpoint inhibitors, underscoring the clinical need to unravel mechanisms that regulate immune cell evasion to overcome treatment resistance by combinatorial treatment approaches[25]. Previous studies demonstrated that during tumor development lung tumors present sparse infiltration of T cells that is associated with a waning of neoantigen presentation and a loss of clonal neoantigens suggesting immune-editing with tumor progression[26]. Mechanistically, cancer cells acquire the capability to inactivate genes regulating IFNγ receptor signaling and to inhibit the expression of major histocompatibility complex (MHC) molecules[27]. In SCLC, particularly epigenetic regulation of MHC-I expression has been described[28,29], which is linked to neuroendocrine and non-neuroendocrine cell states in SCLC, characterized by NOTCH activation level[30] and CD44 expression[31]. Response to immune checkpoint blockade have been previously annotated to reduced neuroendocrine features and NOTCH pathway activation triggered by Lysine-specific demethylase 1a (LSD1) inhibition[32], whereby NOTCH activation was in line with MHC-I upregulation[33]. Our data shows that ERBB2 inhibition affected the antigen-presenting and processing machinery including peptide transporters as TAP1 and TAP2[34] and directly the expression of MHC-I, but did not rise evidence to regulate neuroendocrine features of SCLC cells.

Supporting our kinase inhibition approach, studies described a reduced antigenicity in oncogene-kinase-driven cancers[35]. In line with these studies, we found that ERBB2 signaling via AKT and MAPK pathways suppresses MHC-I expression in SCLC. In contrast, we show that ERBB2 blockade induces phosphorylation of TBK1 and STING signaling that resulted in an increase of MHC-I expression. Most strikingly, selective inhibition of STING attenuates MHC-I expression induced by ERBB2 blockade in SCLC cells which is in line with previous studies describing enhancement of cytosolic DNA priming and recruitment of STING upon inhibition of ERBB2[36].

In summary, we demonstrate that the ERBB2 signaling axis regulates MHC-I expression and immune evasion in SCLC. Selective targeted blockade of the ERBB2 signaling axis was sufficient to induce MHC-I expression and to prevent immune evasion in autochthonous murine SCLC. Most strikingly, we demonstrate a synergistic effect of ERBB2 and anti-PD-1 targeted treatments that elicit profound responses in preclinical SCLC models, suggesting this combination for future clinical trials in patients with SCLC.

## Methods
### Animal experiments

All mice were housed in climate-controlled rooms with a minimum air exchange rate of eight times per hour. The ambient temperature was maintained between 20–24 °C and the relative humidity ranged from 45–65%. An automated 12:12 h light-dark cycle was implemented in all animal holding rooms.

The genetically engineered SCLC mouse model is driven by conditional deletion of the tumor suppressor genes *Rb1* and *Trp53*[37]. Male and female mice with a C57BL/6 background, a minimal age of 6 weeks and a minimal weight of 20 g are included in the study. Animals of both sexes were randomly assigned to experimental groups and sex was not considered in the study design or analysis. Previous to intratracheal Adeno-Cre virus application, mice received anaesthesia by intraperitoneal injection of xylazine (Bayer AG, Germany)/ketamine (Zoetis Inc., USA) (10/100 mg/kg/KGW; max. injection volume 0.1 ml/10 g mice). The University of Iowa Viral Vector Core (http://www.medicine.uiowa.edu/vectorcore) provided the viral vectors. Tumor development was monitored by μCT (LaTheta mCT, Hitachi Alcoa Medical, Ltd). Treatment regimens started upon the identification of a measurable target lesion with a diameter of ≥1 mm. Mice were randomly assigned to one of four therapy groups, ensuring a similar distribution of target lesion diameters at the beginning of treatment. The cohorts encompassed four therapy groups, with all treatments administered every three days as follows: (Group 1) vehicle (phosphate-buffered saline; PBS); (Group 2) anti-PD-1 (Bio X Cell, Cat. # BP0146, clone RMP1-14) with 10 mg/kg BW; (Group 3) ERBB2i mubritinib (Selleck, Cat. #S2216) or lapatinib (Selleck, Cat. #S2111) with 10 mg/kg BW at a concentration of BW via oral application; (Group 4) combination therapy of anti-PD-1, ERBB2i. Under isoflurane anaesthesia tumors were assessed in serial μCT scan on a weekly basis. Tumor response and progress during treatment were classified according to mouse-adapted RECIST criteria v1.1, with a slice thickness of 0.3 mm.

To evaluate local tumor growth and metastatic spread, orthotopic injections of ERBB2 KO and MHC-I KO in comparison to the WT were performed. Immunocompetent C57BL/6 J mice of both genders with a minimum weight of 20 g were used. Mice were anesthetized via intraperitoneal injection of xylazine/ketamine (10/100 mg/kg/KGW; max. injection volume 0.1 ml/10 g mouse). After confirming surgical tolerance, the right thoracic area was shaved and disinfected. A total of $5 \times 10^6$ tumor cells in 80 μL PBS were injected into the right thoracic cavity, targeting the fourth or fifth intercostal space. Tumor growth was monitored by serial in vivo microCT imaging (LaTheta mCT, Hitachi Aloka Medical Ltd.) under isoflurane anaesthesia. To investigate metastatic potential of MHC-I KO and ERBB2 KO in comparison to WT $1 \times 10^6$ tumor cells were injected into the tail vein. C57BL/6 J mice with a minimum weight of 20 g were used for immunocompetent experiments, while NOD scid gamma mice (NSG) were used for immunodeficient settings. Both male and female mice were included. The progression of tumor growth and the occurrence of metastases were continuously monitored through MRI scans employing a 3.0 T MRI system from Philips, with a specialized small animal coil (40 mm in diameter) from Philips Research. For the comparison between ERBB2 KO and WT, MRI slices were evaluated by quantifying metastases per MRI slice, using a size threshold of ≥1 mm in diameter. To examine immune cell involvement in SCLC tumor control, $5 \times 10^6$ ERBB2 KO cells were injected subcutaneously in both femoral flanks (maximum 0.1 mL per site) in isoflurane anesthetized mice. Mice were randomly assigned to either control or immune cell depletion group, which received CD4 (clone GK1.5, BioXCell, Cat. #BE0003-1) and CD8a (clone 2.43, BioXCell, Cat. #BE0061) depleting antibodies (200 μg/mouse,

diluted in PBS) twice per week with beginning two days after sub-cutaneous tumor cell injection.

For comparison of growth behavior in an immunodeficient set-ting, ERBB2 KO and WT cells were injected subcutaneously into NSG mice. Tumor development was continuously monitored by caliper measurements. Experiments were terminated before subcutaneous tumors exceeded 1.5 cm in diameter. The maximal tumor burden permitted by the ethics committee was not exceeded. For orthotopic lung tumors, tumor growth and metastases were monitored in CT/MRI and experiments ended at the first detection of distress or endpoint criteria. For the intravenous metastasis model, hepatic lesions were monitored via MRI and experiments were terminated at the first detection of lesions ≥1 mm or upon reaching predefined humane endpoints.

## Immunohistochemistry

Human primary SCLC and metastases were diagnosed by trained lung pathologists based on histological examination of formalin-fixed paraffin-embedded (FFPE) material in routine diagnostics. Pri-mary antibodies against MHC-I (abcam, clone EPR1394Y, Cat. # ab134189), ERBB2 (Roche, clone 4B5, Cat. # 790-2991) and NCAM-1 (Zytomed, Cat. #RBK050) were used. All primary antibodies used were validated for specificity and performance by reference to the manufacturer's datasheets, previously published studies and by in-house testing. Murine primary SCLC and metastasis were detected in harvested organs by hematoxylin and eosin (H&E) staining. Samples were fixed in 4 % PBS-buffered formalin and paraffin embedded. The tissues, cut into 3 μm sections, underwent deparaffinization and staining using the LabVision Autostainer-480S (Thermo Fisher Sci-entific). Secondary antibodies were purchased from ImmunoLogic (BrightVision + ). Clinicopathologic characteristics have been col-lected for matched and non-matched primary and metastases sam-ples (Supplementary Data 10). MHC-I and ERBB2 expression were evaluated using the image analysis software QuPath (v.0.5.0), applying a semi-quantitative approach. Digital images of the stained slides were generated using the Nanozoomer S360 Digital slide scanner. The H-score for MHC-I was calculated by QuPath based on the intensity and proportion of positive tumor cells, using the for-mula H-score = $(1 \times \%weak) + (2 \times \%moderate) + (3 \times \%strong)$.

## Flow cytometry

Tumor tissue was isolated by 40 μm cell strainer and red blood cells were lysed by ACK Lysing Buffer. After washing with PBS, the cell suspension was incubated with the staining mix for 30 min at 4 °C. The following antibodies and isotype controls were used for staining: CD3 (Alexa-Fluor-700, clone 17A2, Biolegend, Cat. #100216), CD4 (PE-Daz-zle 594, clone GK1.5, Biolegend, Cat. # 100456), CD45 (APC-Cy7, clone 30-F11, Biolegend, Cat. # 103116), CTLA-4 (PE, UC10-4B9 Thermo Fisher, Cat. #14-1522-82), CD56 (APC, R&D Systems, clone 809220, Cat. # FAB7820A), CD8a (FITC, clone 53-6.7, Biolegend, Cat. # 100705; Pacific blue, clone 53-6.7, Biolegend, Cat. # 100728), H2Kb (Pacific Blue, clone AF6-88.5, Biolegend, Cat. # 116517), PD-1 (APC, clone 29 F.1A12, Biolegend, Cat. # 135210), PD-L1 (PE-Cy7, clone 10 F.9G2, Biolegend, Cat. # 124313), TIM-3 (PerCP-Cy5.5, clone B8.2C12, Biole-gend, Cat. # 134011), Rat IgG2aK (Biolegend FITC, Cat. # 400505, PE, Cat. # 400507, PerCP-Cy5.5, Cat. # 400531, APC, Cat. # 400511, Alexa Fluor 700, Cat. # 400528), PE-Dazzle594 Armenian Hamster IgG (PE-Dazzle594, clone HTK888, Biolegend, Cat. # 400951), Rat IgG2bK (PE-Cy7, clone RTK4530, Biolegend, Cat. # 400617) and mouse BALB/c IgG2aK (Pacific Blue, clone G155-178, BD Biosciences, Cat. # 558118). As viability dye the Zombie Aqua Fixable Viability Kit (BioLegend, Cat. # 423101) was applied. The data were analyzed using Kaluza Software (v.2.1, Beckmann Coulter). Apoptosis and necrosis were analyzed by simultaneous detection of cell surface Annexin V and propidium iodide (Biolegend, Cat. # 640914) as described by the manufacturer.

## Cell culture

Human lung cancer cell line Glc8 was kindly provided by Roman K. Thomas (Department of Translational Genomics, University of Cologne, Germany) and Reinhard Büttner (Institute for Pathology, University Hospital Cologne, Germany). Additionally, human SCLC cell lines NCI-H82 (ATCC, Cat. # HTB-175), NCI-H69 (ATCC, Cat. # HTB-119), NCI-H196 (ATCC, Cat. # CRL-5823), NCI-H735 (ATCC, Cat. # CRL-5978), NCI-H1688 (ATCC, Cat. # CCL-257) were used. Murine SCLC cell lines were established after harvesting primary SCLC lung tumors or liver metastasis of the autochthonous SCLC mouse model. Tumor tissue was mechanically dissociated using 40 μm cell strainers (BD Falcon) and ACK lysis buffer. Isolated cells were cultivated in RPMI medium (Life Technologies) with 10% FBS (Sigma-Aldrich) and 1% antibiotics (peni-cillin/streptomycin, Life Technologies) and regularly screened for mycoplasma contamination and confirmed negative by PCR. After 5 passages, the cell lines were defined as stable and were used for functional experiments. mSCLC P1, mSCLC P2, mSCLC P3, mSCLC P4 and mSCLC P5 were derived from primary tumor, mSCLC M1 from metastasis. Cell authentication was performed by PCR genotyping and regular assessment of neuroendocrine marker expression, which was routinely tested every 2 weeks and before experimental use. When specified, mubritinib, lapatinib (Selleckchem) and neratinib (Med-ChemExpress, Cat. # HY-32721) as ERBB2 inhibitor treatment, IFN-gamma (IFNγ) and IFN-beta (IFNβ) (PeproTech) or SN-011 (Sell-eckchem) as STING inhibitor treatment were performed at the indi-cated concentration.

## CRISPR-Cas9

In murine derived SCLC tumor cells, Cas9-mediated knockout (KO) of B2M and ERBB2 was achieved applying the Santa Cruz CRISPR con-structs according to manufacturer's advice. Twenty-four hour prior to transfection, cells were seeded in 6 well plates at ~60% confluency. They were transfected with Ultracruz Transfection reagent (Santa Cruz, Cat. # sc395739) and the Plasmids containing Cas9 and the corresponding guide RNA (B2M Santa cruz, Cat. # sc-419281-nic or ERBB2 Santa cruz, Cat. # sc-420219-nic-2). After 48 h post-transfection, puromycin (3 μg/ml) was applied for selection of successfully trans-fected cells, single clones were plate-sorted. The efficiency of the knock-out was validated with western blot and flow cytometry.

## RNA sequencing

Cells were seeded in a 6-well plate and exposed to different experi-mental conditions for 48 h. RNA was extracted using Qiagen RNeasy Mini Kit, according to the manufacturer's protocol (Qiagen, German-town, MD, USA). Corresponding to the manufacturer's requirements, a concentration of 100–200 ng/μL was used from each sample. Libraries of 3'mRNA were obtained from total RNA using the Lexogen QuantSeq kit according to standard protocol. After validation and quantification (2200 TapeStation, Agilent Technologies, Santa Clara, CA, USA and Qubit System, Invitrogen, Carlsbad, California, CA, USA respectively), pools of cDNA libraries were generated. Pools were quantified using the KAPA Library Quantification kit (Peqlab, Radnor, PA, USA) and the 7900HT Sequence Detection System (Applied Biosystems, Foster City, PA, USA) and lastly sequenced on an Illumina HiSeq4000 or NovaSeq6000 sequencer using a 2× 100 base pair protocol.

In brief, the FASTQ files were aligned to the ensembl GRCm39 reference using Bwa v0.7.17 and Samtools v1.13. Duplicate reads were marked with Picard v2.26.0. Mutations were called using GATK Mutect2 v4.2.1.0 in tumor only mode. The panel of normal for this analysis included 14 healthy murine samples from our previous study[38]. The mutations were filtered using GATK FilterMutectCalls (v4.2.1.0) and annotated using ensemble-vep v113.4. Variants were excluded if present in the strain reference files from Wellcome Sanger Mouse Genome Project (mgp_REL2021_indels and mgp_REL2021_snps) or in the ensembl variation database v113.4 or if they were predicted not to

affect protein sequence. Exon coverage was collected using GATK CollectReadCounts (v.4.2.1.0) and normalized to reads/million. The total normalized coverage within ERBB2 exons was then plotted. The expected coverage at copy number 2 was estimated using the median ERBB2 coverage in healthy samples.

## Whole exome sequencing

1 μg of DNA was fragmented using ultrasonic treatment (Covaris, Inc., Woburn, MA, USA). The resulting fragments were end-repaired, and adapters were ligated. Library preparation was performed using the Agilent SureSelectXT HS2 Mouse All Exon kit. Sequencing was conducted on an Illumina NovaSeq 6000 instrument using a paired-end 2 × 100 bp protocol, with a target coverage of 250×.

## Western blot

Cells were seeded in a 6-well plate and exposed to different experimental conditions for 24 to 72 h. Cells were then washed in PBS and lysed in RIPA buffer (Cell Lysis Buffer (Cell Signaling), NaF 10 mM, PMSF 1 mM) containing phosphatases and proteases inhibitors. The BCA Protein Assay (Pierce, Thermo Fisher Scientific Inc., USA) was used to determine protein concentrations. To separate the protein samples via SDS-PAGE, the cell lysates were incubated with 4× NuPage® LDS buffer and sample reducing agent (10×) (Thermo Fisher Scientific Inc., USA) for 10 min at 80 °C. The samples were loaded onto NuPAGE Bis-Tris Gels 4–12% (Thermo Fisher Scientific Inc., USA). Protein transfer to a nitrocellulose membrane (Amersham Hybond-C Extra) was performed by wet blotting. To perform immunodetection, membranes were blocked with 5% (w/v) BSA diluted in Tris Buffer Saline with Tween 0.05% (TBS-T; pH 8), incubated with primary antibodies (pERBB2, Cell signaling, Cat. #2243; pTBK1, Cell signaling, Cat. #5483; pERK1/2, Cell signaling Cat. #9106; pAKT, Cell signaling, Cat. #9271; ERBB2, Cell signaling, Cat. #4290; TBK1, Cell signaling, Cat. #3504; ERK1/2, Cell signaling; Cat. #9102; AKT, Cell signaling, Cat. #9272; Rig-I (D14G6), Cell signaling, Cat. #3743; cGas (D3080), Cell signaling, Cat. #31659; pSting, Invitrogen, Cat. #PA5-105674; Actin (Clone C4), MP, Cat. #691001) and HRP-coupled anti-mouse/rabbit secondary (Millipore) were used. Immunodetection was performed using Clarity Western ECL Substrate (Bio-Rad, Hercules, CA, USA) and ChemiDoc MP Imaging System (Bio-Rad, Hercules, CA, USA). Fiji software was used to determine and analyze densitometric profiles (v.1.53q; US National Health Institute, USA). Each Western blot experiment was repeated at least 3 times under the same conditions in order to provide a representative protein expression profile. Unprocessed blots are available as Supplementary Information file. The RTK assay (R&D Systems) was conducted using 300 μg of protein.

## Mass spectrometry

Cells were seeded in a 10-cm dish and exposed to different experimental conditions for 48 h (mubritinib, lapatinib, neratinib 1 μM respectively). Cells were washed in PBS and lysed in Urea buffer (8 M Urea (Sigma-Aldrich) in 50 mM TEAB (Sigma-Aldrich)). The chromatin is degraded using a Bioruptor (10 min, cycle 30/30 s). The BCA Protein Assay (Pierce, Thermo Fisher Scientific Inc., USA) was used to determine protein concentrations. The cells were incubated 1 h with DTT (Applichem) 5 mM then 30 min with CAA (Merck) 40 mM. Samples were diluted with TEAB 50 mM to achieve a final concentration of Urea ≤ 2 M. Samples were digested with trypsin (Serva) at an enzyme:substrate ratio of 1:75 and incubated at 25 °C overnight. The enzymatic digestion was stopped by addition of a 1% solution of formic acid (Honeywell/FLUKA). 150 μL of suspended peptide samples were applied to the equilibrated TiO2 Spin Tip (Thermo Scientific). After centrifugation, the samples were re-applied to the Spin Tip in the microcentrifuge tube. The columns were washed by adding 20 μL of Binding/Equilibration Buffer and then washed by adding 20 μL of Wash Buffer. Finally, the columns were washed by adding 20 μL of LC-MS grade water (Merck). Excess liquid was removed by blotting the bottom of the spin tip onto a clean laboratory tissue and 50 μL of phosphopeptide elution buffer (Fischer Scientific) was added to the spin tip. The eluates were then immediately dried in a high-speed vacuum concentrator to remove the phosphopeptide elution buffer. The eluates were suspended in 50 μL of 0.1% formic acid for peptide concentration measurements using the Pierce™ Quantitative Colorimetric Peptide Assay Kit or direct MS analysis. Samples were analyzed by the Cellular Stress Responses in Aging-Associated Diseases (CECAD) Proteomics Core Facility (University of Cologne, Germany) on an Orbitrap Exploris 480 mass spectrometer (Thermo Scientific) coupled to an Evosep ONE (Evosep). The Evosep was run with its Whisper Zoom 20 SPD gradient using an Aurora Elite pulled-tip column (Ionopticks). The mass spectrometer was operated using a WHISH-DIA approach[39]. MS2 spectra were acquired in the range of 400 to 1000 m/z at 60k resolution in 25 m/z windows, resulting in 24 scans total. Fragments were acquired in a range of 250 to 1500 m/z with a normalized AGC target of 1000% and 30% normalized HCD collision energy. Every 6 scans, an MS1 scan was inserted, which was acquired at a resolution of 120k in the range of 390–1010 m/z. Samples were analyzed in Spectronaut 19 (Biognosys) using standard setting for directDIA analysis, but quantification performed on MS1 level and requiring at least 6 fragment ions. Results were searched against the canonical murine Uniprot reference proteome (UP589, downloaded 15/01/2025) with follow-up analysis performed in Perseus 1.6.15[40].

For proteome profiling performed with the DKTK Proteomics Core Facility (Goethe University Frankfurt, Germany), cell pellets were lysed in urea lysis buffer (8 M urea, 20 mM HEPES, pH 8.0, 1 mM sodium orthovanadate, 2.5 mM sodium pyrophosphate, 1 mM beta-glycerophosphate). Protein concentrations of the lysates were determined using the 660 nm assay kit (Thermo Fisher Scientific) according to the manufacturer's instructions. 800 μg protein per sample were reduced with DTT (10 mM for 1 h at 37 °C), alkylated with iodoacetamide (25 mM for 15 min at 37 °C in the dark) and digested using Lys-C (Wako/Fujifilm) for 2 h at 37 °C in an enzyme-to-substrate ratio of 1:50 (w/w). After dilution with 20 mM HEPES (pH 8.0) to a concentration of 2 M urea, digestion was continued overnight with trypsin (Promega) at 37 °C and 1:50 (w/w) enzyme-to-substrate ratio. The peptide mixtures were acidified and purified using C18 spin tips (Havard). For global proteome analysis, 10 μg peptide were dried by vacuum centrifugation and then dissolved in 0.1% formic acid (FA). Peptide concentrations were determined using a fluorometric peptide assay (Thermo Fisher Scientific). To enrich phosphopeptides, 400 μg peptides were bound to TiO₂ columns using the High-Select TiO₂ Phosphopeptide Enrichment Kit (Thermo Fisher Scientific). Collected phosphopeptides from the eluate were also dried and resuspended in 0.1% FA. The peptide samples were analyzed by LC-MS/MS on a Vanquish Neo UHPLC system (Thermo Fisher Scientific) coupled online to an Orbitrap Astral mass spectrometer (Thermo Fisher Scientific) in a data-independent acquisition scheme (DIA). 400 ng of peptides from each sample were concentrated and desalted on a PepMap Neo trap cartridge (Thermo Fisher Scientific, particle size 100 Å, inner diameter 300 μm, length 5 mm), followed by separation on a 15 cm PepMapNeo analytical column (Thermo Fisher Scientific) using a 30 min method (27 min linear gradient) of 1% to 28% acetonitrile in 0.1% formic acid at a flow rate of 800 nl/min. Precursor ion survey scans were acquired using the Orbitrap mass analyzer with the following parameters: resolution 240,000, scan range m/z 380–980, automatic gain control (AGC) target $5 \times 10^6$, maximum injection time 10 ms, RF lens setting 40%. For fragment ion scans using the Astral mass analyzer, precursor ions were isolated for collision-induced dissociation (HCD) through each survey scan with an isolation window of m/z 2, resulting in 299 scan events. The normalized HCD collision energy was set to 25% and for

fragment ion analysis the AGC target was $5 \times 10^4$ at a maximum injection time of 3 ms.

Raw DIA data were analyzed using Proteome Discoverer (v.3.1.1.93, Thermo Fisher Scientific). Spectra were searched against the Uniprot mouse reference proteome and 245 frequently observed contaminants using the CHIMERYS search algorithm. The mass tolerance for fragment ions was set to 10 ppm. Oxidation of methionine was considered as dynamic modification while carbamidomethylation of cysteine was defined as a fixed modification. The peptide length was defined as between seven to 30 amino acids with one allowed missed cleavage site. One to four charges per peptide were allowed. At both peptide and protein level, the false discovery rate (FDR) was set at 1%. For phosphoproteome analysis, Phospho Modifications (S, T, Y) were set as dynamic modification. Further data processing was done using R studio (v.2024.09.1). First, contaminants were removed. To control for equal sample loading, intensities from each LC-MSMS run were normalized on the median of the summed-up intensities from each sample[41]. Phosphoproteome analysis was performed at the site-specific level. Peptide abundances were merged by sequence with the modification sites. If more than one peptide sequence group matched a site (e.g. due to miscleavages), the abundance of all peptide sequence groups for the site was summed.

## T cell cytotoxicity assays

Tumor cells were 48 h incubated with RPMI medium for the control group or 10 nM Mubritinib diluted in RPMI medium and washed with PBS. For pulsing tumor cells were incubated with 10 µL/10,000 cells AIM-V medium with 10 µg/mL of OVA (Anaspec, Inc.) for 1 h at 37 °C. Afterwards, the cells were washed with RPMI medium. 35,000 cells/well were seeded into a 12-well plate. For T cell preparation OT-I splenocytes were isolated according to the protocol of the Pan T Cell Isolation Kit II mouse (Miltenyi Biotec). After washing, the T cells were counted and added to the well plate in different effector/tumor cell ratios (E:T 1:1, 2:1, 4:1, 8:1) and incubated at 37 °C, 5% CO$_2$ for 24 h.

## TCR sequencing

150 µL blood was drawn from treated mice in heparin coated tubes. Red blood cells were lysed by treating the sample with 850 µl ACK lysis buffer for 10 min at RT followed by washing with PBS. Cells were then resuspended in MACS buffer and negatively selected for CD3 using the Pan T Cell Isolation Kit II, mouse (Miltenyi Biotec, Germany). 10% of the isolated T cells were used to check sample purity and viability by flow cytometry using Aqua Zombie and CD3, CD4 and CD8 stains. 90% of the isolated T cells were analyzed by TCR sequencing in the Cologne Center for Genomics using. FastQC software was used to evaluate the quality of the returned sequencing data. The CellRanger analysis pipeline from 10× Genomics was used to perform the processing of raw FASTQ files for standard bioinformatic analysis.

Publicly available data from a healthy C57BL/6 J mouse by 10× Genomics were used as reference data. The dataset is listed as "PBMCs from C57BL/6 mice (v1)" dataset and was analyzed in the same manner as our samples. Cell clustering and visualization in 2 dimensions (UMAP), annotation of cell types, cell type differential gene expression analysis and visualization was done with Loupe browser from 10× Genomics (v.8.0.0). For clonotype distribution the V(D)J browser (v.5.1.0) from 10× Genomics was used. Clonality was calculated with the clonality function from the LymphoSeq R package and Morisita's overlap index with the R package tcR. Network plots were generated by the networkD3 package and calculated with Biostrings. All statistical analyses were performed using R (v.4.2.0) in its integrated development environment RStudio (v.2023.06.1 + 524).

## Single cell RNA sequencing analysis

Single-cell RNA sequencing of tumor tissue was performed by Singleron Biotechnologies GmbH (Cologne, Germany). Murine SCLC tumor samples were harvested from lungs of treated mice, washed in PBS, covered by Sample Preparation Buffer (Singleron Biotechnologies) and shipped on ice. Samples were processed within 72 h after tumor isolation using sCelLiVE™ Tissue Dissociation Buffer (Singleron Biotechnologies) and microfluidic SCOPE-chip™. Barcode hybridization was followed by reverse transcription and cDNA amplification. Amplified cDNA was fragmented, ligated to adapters and PCR amplified to construct a sequencing library suitable for Illumina based sequencing. For single-cell transcriptome analysis, quality control was performed prior to downstream analysis. CeleScope was used to generate a single-cell gene expression matrix file based on the raw sequencing data. Standard single-cell gene expression QC metrics were used to identify high quality single cells removing doublets (large gene or Unique Molecular Identifier−UMI counts), dying cells (measuring mitochondrial RNA reads) and debris cells (with small UMI counts). For quality control, cell clustering, cell type annotation, differential gene expression analysis, and visualization, we utilized Scanpy (v.1.9.6)[41], a Python toolkit designed for single-cell data analysis. Following best practices, cells with high mitochondrial or ribosomal gene content were excluded. Specifically, cells were marked as outliers and removed if their mitochondrial gene content exceeded three median absolute deviations (MADs) from the median. Additionally, cells with mitochondrial counts exceeding 8% were filtered out. A threshold of five MADs was applied to other quality control covariates, including total counts (log1p_total_counts), the cumulative percentage of counts from the top 20 expressed genes in a cell (pct_counts_in_top_20_genes), and the number of detected genes (log1p_n_genes_by_counts). Cells with more than 150 detected genes were retained for downstream analysis. Potential doublets were detected and removed using SOLO, a semi-supervised deep learning method[42] implemented via scvi-tools (v.1.0.4). The expression matrix was globally scaled by normalizing each gene's expression relative to the total expression per cell. Subsequently, values were multiplied by a scaling factor of 10,000 and transformed using a natural logarithm with a pseudocount of 1. A lower dimensional representation of the data was obtained using UMAP (Uniform Manifold Approximation and Projection)[43], a nonlinear manifold learning approach. Prior to cell type annotation, cells were clustered using the PhenoGraph-Louvain[44] algorithm implemented in Scanpy. Analyses were conducted in Python (v.3.9), and plots were generated and visualized with Scanpy.

## Analysis of public RNA sequencing and scRNA-seq data

Public single-cell RNA-seq data from 19 SCLC patients with 21 Biospecimens published by Chan and colleagues[10] were obtained from the CELLxGENE platform. We downloaded the pre-annotated h5ad object and performed downstream analyses with Scanpy (v.1.9.6) and Seurat (v.5.0.3). No additional pre-processing was applied to the data and metadata annotations provided in the original dataset were used.

Processed RNA sequencing data from IMpower133[3] (n = 271) and relevant clinical data were obtained from the European-Genome-phenome Archive under the identifier EGAS50000000138. Subgroup analysis has been performed based on treatment arms and TMB. The cutoff was determined using Cutoff Finder[44].

## Statistics

Data were analyzed using GraphPad Prism 8 software and figures were prepared with Inkscape software (v.0.92.4). Normality of the data was assessed using the Shapiro−Wilk test. For normally distributed data, statistical differences between groups were evaluated using Student's $t$-test. For data that did not meet the normality assumption, the non-parametric Mann−Whitney $U$ test was applied. For multiple comparisons, unpaired two-tailed $t$-tests were performed with Bonferroni correction applied to adjust for multiple testing. Pearson correlation coefficients were calculated using GraphPad Prism (v10.6.1) to assess linear relationships between

variables. The correlation coefficient $r$ is reported. $p$-values < 0.05 were considered to be statistically significant.

## Data analysis and reproducibility

All key experiments were independently repeated at least three times using biological replicates. Technical replicates were included, where appropriate, for example in flow cytometry-based quantifications. Experimental results were consistent across replicates. Any variation observed was within the expected biological or technical range and did not affect the interpretation of the findings. No experiments failed to replicate under the conditions described. Investigators were not blinded to group allocation during experiments or outcome assessment. Experimental procedures and data analyses were performed using standardized protocols and automated or semi-automated quantification methods to minimize potential bias. No formal sample size calculation was performed for any experiments, except for animal studies. Here sample sizes were chosen based on previous experience and published literature and were considered sufficient to detect relevant differences. The number of animals per experimental group was determined based on a statistical power analysis to ensure sufficient statistical robustness. Effect sizes were estimated from previous experience. Sample size calculation was performed using G*Power (v.3.1.9.7) for a two-sided $t$-test with a significance level of 0.05, a power of 0.80, and an effect size of 0.90. Mice were randomized into therapy cohorts prior to tumor induction. Target lesion diameters at the start of treatment were similar across all groups, ensuring balanced baseline conditions. Data from mice were excluded from analysis if the animals died or were euthanized for reasons unrelated to the experimental intervention. Only data from mice that completed the experimental protocol were included in the final analyses.

## Ethics statement

All human subject research was performed in strict accordance with approved protocols by the local ethics committee of the Medical Faculty of the University of Cologne and with the recognized ethical guidelines of the Declaration of Helsinki. Tumor tissue (reference no. 13-091) was obtained during routine clinical procedures from lung cancer patients providing written informed consent. Animal Experiments were performed in accordance with FELASA recommendations. The protocol was approved by the local animal welfare committee of the University of Cologne and authorized by the Landesamt für Natur, Umwelt und Verbraucherschutz (LANUV NRW, Düsseldorf) (TVA 2025-219; 84-02.04.2015.A199; 81-02.04.2020.A026; 81-02.04.2020.A219; 81-02.04.2020.A328). All procedures were conducted in accordance with institutional, national, and international guidelines.

## Reporting summary

Further information on research design is available in the Nature Portfolio Reporting Summary linked to this article.

## Data availability

Sequencing and proteomic data generated in this study are publicly available in the following MINSEQE-compliant repositories: Proteomic data are available in the PRIDE database (ID PXD065735) and in the MassIVE repository (ID PXD066359). Whole-exome sequencing data have been deposited in the Sequence Read Archive (SRA) (BioProject ID PRJNA1293554). TCR sequencing data are available in the Gene Expression Omnibus (GEO) under accession number GSE282715. Bulk RNA sequencing data are available in GEO under GSE283573 and GSE303491, and single-cell RNA sequencing data under GSE283827. Additional materials are available upon request from the corresponding authors. The single-cell RNA-seq data from Chan et al.[10] used for comparative analysis were accessed via Cell × Gene [https://cellxgene.cziscience.com/e/34deb33b-a50e-4993-a38b-1c0e5079c1c2.cxg]. RNA expression data from human SCLC cell lines were obtained from the GDSC-MGH-Sanger dataset accessed via CellMinerCDB [https://discover.nci.nih.gov/cellminercdb/]. The IMpower133 dataset was accessed from the European Genome-phenome Archive under the identifier EGAS50000000138 via https://ega-archive.org/ with the approval DA01145. These data were not generated in this study. The remaining data are available within the Article, Supplementary Information or Source Data file. Source data are provided with this paper.

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

## Acknowledgements

Human lung cancer cell lines were kindly provided by Roman K. Thomas (Department of Translational Genomics, University of Cologne, Germany) and Reinhard Büttner (Institute for Pathology, University Hospital Cologne, Germany). We thank Renato Liguori (Bioinformatics and Computational Pathology, Friedrich-Alexander Universität Erlangen-Nürnberg, Germany) and Yussuf Hajjaj (Chair for Experimental Medicine 1, Friedrich-Alexander Universität Erlangen-Nürnberg, Germany) for handling the IMpower133 data under DA01145, Theodoros Georgomanolis (Cologne Excellence Cluster on Aging and Aging-Associated Diseases, University of Cologne, Germany) for his bioinformatics support regarding TCR sequencing, Jaewon Kwak (Institute for Pathology, Faculty of Medicine and University Hospital of Cologne, Germany) for the support in patient data management, Johannes Lindemeyer (Department of Diagnostic and Interventional Radiology, Faculty of Medicine and University Hospital of Cologne, Germany) and the Core Facility Experimental and Preclinical Imaging Cologne (University of Cologne, Faculty of Medicine and University Hospital Cologne, Germany) and Julian Kallinowski (Department I of Internal Medicine, Faculty of Medicine and University Hospital of Cologne, Germany) for support with small animal imaging. We thank Kerstin Wennhold (Center for Molecular Medicine Cologne, Faculty of Medicine and University Hospital of Cologne, Germany) for sharing the T cell OVA cytotoxicity assay protocol. We thank the CECAD Proteomics Core Facility and the DKTK for proteomics analysis. Proteomics data acquisition at the University of Cologne was performed on an instrument granted by the Deutsche Forschungsgemeinschaft under grant INST 216/1163-1 FUGG. We thank Martine Pape and Marion Bodach of DKTK proteomics platform Frankfurt for their excellent technical assistance. We thank the Cologne Center for Genomics (CCG) for performing whole-exome and RNA sequencing and their support in data generation. The figures were created with BioRender.com: Meder, L. (2025) https://BioRender.com/z2ehzfh, Meder, L. (2025) https://BioRender.com/su67ybp, Meder, L. (2025) https://BioRender.com/4u5k5v3, Meder, L. (2025) https://BioRender.com/w6kb2vb, Meder, L. (2025) https://BioRender.com/dwd4xgv, Meder, L. (2025) https://BioRender.com/3wkefcf, Meder, L. (2025) https://BioRender.com/x5oxh7c, Meder, L. (2025) https://BioRender.com/bxcwg6c, Meder, L. (2025) https://BioRender.com/vkn5pfz, Meder, L. (2025) https://BioRender.com/y4lktuu, Meder, L. (2025) https://BioRender.com/eyhkfjh, Meder, L. (2025) https://BioRender.com/a7q733t. This work was supported by Wilhelm Sander Stiftung, Grant No.: R2024.171.1 (RTU), Interdisciplinary Center for Clinical Research of the Faculty of Medicine of the Friedrich-Alexander University Erlangen-Nürnberg Grant No.: 29133011 (LM), German federal state North Rhine Westphalia; Grant No.: 1411ng005 (RTU); Grant No.: LS-1-1-030a (HCR); the CANTAR network Grant No.: NW21-062B funded through the program "Netzwerke 2021", an initiative of the Ministry of Culture and Science of the State of North Rhine Westphalia, Germany (JB), German Cancer Aid; Grant No.: 70113009 (RTU); Grant No.: 111724 (HCR); Grant No.: 70113307 (LM; DS; JB; FB), Grant No.: 70114593 (KN) Fritz-Thyssen Foundation; Grant No.: 10.21.1.026MN (LM), German Research Foundation; Grant No.: UL379/1-1 (RTU); CRC-1399/A01/C02 (HCR; FB), CRC-1399/ project ID 413326622 (JB), CRC1310/project ID 325931972

(CK, FK, JB), CRC-1530/B05 (RTU), GRK 3110/1 (KN; JW; RB; RTU), CRC-1530/INF (NA), Grant No.: 497777992 (NA), Else Kröner-Fresenius Foundation; Grant No.: EKFS-2014-A06 (HCR), Memorial Grant 2018_EKMS.35 (JB), and MD Research Stipend of the Else Kröner Forschungskolleg Clonal Evolution in Cancer, University Hospital Cologne, Germany (DS), Ministry of Culture and Science of the German state of North Rhine-Westphalia as part of the support programme Cancer Center Cologne Essen (NA).

## Author contributions

Conceptualization: L.M., H.C.R., N.A., R.T.U. Methodology: L.M., C.I.O., C.L.B., R.G., D.S., C.K., J.W., F.B., R.B., T.O., F.K., J.B., H.C.R., N.A., R.T.U. Investigation: L.M., C.I.O., C.L.B., R.G., C.V.O., D.S., M.K., M.N., I.G.K., K.N., X.Z., L.U., B.H., J.J., M.E., A.F., H.G., F.B., R.B., T.O., N.A., R.T.U. Visualization: L.M., C.I.O., C.L.B., M.K., D.S., R.G., N.A. Funding acquisition: L.M., J.B., D.S., F.B., H.C.R., R.T.U. Project administration: L.M., R.T.U. Supervision: L.M., R.T.U. Writing— original draft: L-.M., C.I.O., C.L.B., R.G., C.K., R.T.U. Writing—review & editing: L-.M., C.I.O., C.L.B., R.G., C.V.O., D.S., C.K., M.K., M.N., I.G.K., K.N., X.Z., L.U., B.H., J.J., M.E., A.F., H.G., J.W., F.B., R.B., T.O., F.K., J.B., H.C.R., N.A., R.T.U.

## Funding

## Competing interests

H.C.R. received consulting and lecture fees from Abbvie, Roche, KinSea, Vitis, Cerus, Lilly, Novartis, Takeda, AstraZeneca, Vertex, and Merck. H.C.R. received research funding from AstraZeneca and Gilead Pharmaceuticals. H.C.R. is a co-founder of CDL Therapeutics GmbH. J.B. has received research funding from Bayer and travel grants by Merck and Bicycle Therapeutics outside of the submitted work. T.O. received research funding from Gilead and Merck KGaA, is a consultant/received honoraria for/from Beigene, Roche, Janssen, Merck KGaA, Gilead, Kronos Bio and Abbvie (all not related to this work). The remaining authors declare no conflicting interests. J.B. has received research funding from Bayer outside of the submitted work. The remaining authors declare no competing interests.

## Additional information

[1]Chair of Experimental Medicine 1, Faculty of Medicine, Friedrich-Alexander-Universität Erlangen-Nürnberg (FAU), Erlangen, Germany. [2]Comprehensive Cancer Center Erlangen-EMN (CCC ER-EMN), Universitätsklinikum Erlangen, Erlangen, Germany. [3]University of Cologne, Faculty of Medicine and University Hospital Cologne, Department I Internal Medicine, Cologne, Germany. [4]Center for Molecular Medicine Cologne, Faculty of Medicine and University Hospital Cologne, Cologne, Germany. [5]University of Cologne, Faculty of Medicine and University Hospital Cologne, Mildred Scheel School of Oncology, Cologne, Germany. [6]University of Cologne, Faculty of Medicine and University Hospital Cologne, Department of Translational Genomics, Cologne, Germany. [7]University of Cologne, Faculty of Medicine and University Hospital Cologne, Institute of Virology, Laboratory of Experimental Immunology, Cologne, Germany. [8]University of Cologne, Faculty of Medicine and University Hospital Cologne, Institute of Pathology, Cologne, Germany. [9]Department of Medicine II, Hematology/Oncology, Goethe University, Frankfurt, Germany. [10]German Cancer Research Center (DKFZ) and German Cancer Consortium (DKTK), Heidelberg, Germany. [11]Frankfurt Cancer Institute (FCI), Goethe University, Frankfurt am Main, Germany. [12]German Cancer Consortium (DKTK), partner site Frankfurt/Mainz, a partnership between DKFZ and UCT Frankfurt-Marburg, Germany, Frankfurt am Main, Germany. [13]Institute of Pathology, Charité Universitätsmedizin Berlin, Corporate Member of Freie Universität Berlin, Humbolt-Universität zu Berlin and Berlin Institute of Health, Berlin, Germany. [14]Core Facility Experimental and Preclinical Imaging Cologne, University of Cologne, Faculty of Medicine and University Hospital Cologne, Cologne, Germany. [15]University of Cologne, Faculty of Medicine and University Hospital Cologne, Department of Diagnostic and Interventional Radiology, Cologne, Germany. [16]University of Cologne, Faculty of Medicine and University Hospital Cologne, Center for Integrated Oncology Aachen Bonn Cologne Duesseldorf, Cologne, Germany. [17]Medical Clinic III for Oncology, Hematology, Immune-Oncology and Rheumatology, University Hospital of Bonn, Bonn, Germany. [18]Department of Hematology and Stem Cell Transplantation, West German Cancer Center, University Hospital Essen, Essen, German Cancer Consortium (DKTK), Essen, Germany. [19]Cancer Research Center Cologne Essen (CCCE), Faculty of Medicine and University Hospital Cologne, University of Cologne, Cologne, Germany. [20]These authors contributed equally: Lydia Meder, Charlotte I. Orschel, Cyrielle L. Bouchez. ✉e-mail: lydia.meder@fau.de; roland.ullrich@uk-koeln.de

