## [Transparent Peer Review file · Nature Communications]

ERBB2 signaling drives immune cell evasion and resistance against immunotherapy in small cell lung cancer

Corresponding Author: Professor Roland Ullrich

Version 0:

Reviewer comments:

Reviewer #1

(Remarks to the Author)

In this manuscript, Ullrich et al. investigate a causative role of ERBB2 signaling and MHC-I downregulation, which resulted in immune evasion of SCLC, a highly aggressive and treatment-resistant cancer. Accordingly, genetic or pharmacological blockade of ERBB2 restores MHC-I expression and T cell infiltration, overcomes immune evasion, and reduces SCLC metastasis. Furthermore, combining ERBB2 inhibition with PD-1 blockade elicits promising therapeutic responses in preclinical SCLC models.

Overall, the manuscript addresses an interesting topic in cancer immunology and presents an immune evasion mechanism in SCLC with translational potential. The data are generally convincing and support the conclusions drawn. The manuscript is well-written and presents its findings clearly.

Critics:

1. A more detailed mechanistic explanation of ERBB2 signaling suppresses MHC-I expression is recommended. For example, how does cGAS-STING signaling drive the expression of MHC-I in SCLC to achieve this effect? Is it a direct effect of type I interferons or an indirect impact of IFN-gamma?
2. How is ERBB2 signaling upregulated in these metastatic SCLC specimens through gene amplification, mutation, overexpression, or augmented by overexpression of EGFR, Notch, or Wnt?

Reviewer #2

(Remarks to the Author)

In this manuscript the authors address the need for improved targeted treatment strategies against SCLC. They report decreased expression of MHC-1 in metastatic SCLC using patient samples and that this also occurs in a Rb1/Trp53-depleted SCLC mouse model. Use of the same model confirmed a functional role for MHC-1 in regulating metastasis and identified erbB2 as a mediator of MHC-1 repression. Further work then characterizes the mechanism of erbB2 regulation of MHC-1 and evaluates the potential of anti-erbB2 + anti-PD1 therapy. Although the work is interesting and addresses an important clinical question, the impact of the manuscript is limited by several factors. First, although I understand the limited number of syngeneic models of SCLC, there is an over-reliance on use of the original autochthonous mouse model, and use of cells from related mouse models with additional modifications would strengthen their conclusions. Second, the data from human patients of an inverse relationship between erbB2 and MHC-1 is not convincing. Third, there are concerns regarding the selectivity and effects of the erbB2i, as detailed below. Finally, some of the experiments are poorly described and lack experimental detail, and I'm not convinced by the interpretation of certain experiments, also explained below.

Specific comments

- 1) In Methods for 1b-c it states that clinico-pathological characteristics have been collected for the patient specimens but these are not provided.

2) In 2a-c, metastasis is monitored following intravenous injection. This is not the same as metastasis from the primary site because it skips initial tumour invasion and intravasation into the bloodstream. In the orthotopic model (Ext Data 1a-d) they just look at primary tumour growth. What happens if the primary tumours are resected at the same size and metastasis is measured?

3) Fig 3. It is unclear how many cell lines are being measured, and the identify/nature of these lines. In the legend it states n=4 and n=5 but it is unclear if this is referring to no of cell lines, or biological replicates. If no of cell lines, are these from different mice, or were multiple lines derived from the same tumour? I assume Figs 3a,c,d provide important data but insufficient info is provided to correctly interpret them.

In 3c its unclear what the relevance of overexpressed MAPK1 and MAPK3 is if the phosphorylated form is not enhanced (3d).

Also the MAPKs that are upregulated in the metastatic samples are p38 stress-activated MAPKs ie MAPK13. These are not 'classical' erbB2 effectors and the authors should indicate this.

In 3d the specific regulated phospho-sites are not indicated. This information must be provided because it is unclear whether the changed phosphorylation is actually on important regulatory sites.

4) MS methods are not provided beyond sample prep.

Access to raw files is not provided.

5) Fig 3f.

Where is confirmation that knock-out did not affect SCLC proliferation?

Again, does it indicate independent knock-out lines, or replicates for one line?

Given that the lines are clonal, the authors should confirm that erbB2 rescue reverses the effects on gene expression.

6) 3h why is FACS plot labelled HER2, should be MHC-1

7) Fig 3i, poorly explained. All legend says is 'mean protein expression scores' . How was this experiment done? What is the relevance of phosphorylation of these proteins? Also, these experiments utilized mubritinib as an erbB2i, but this drug also targets the electron transport chain. The authors should confirm mubritinib-induced inhibition of erbB2 in the cells and utilize another approach to confirm that the effect is erbB2-dependent eg knockdown.

8) Fig 3j-k. Again, confirm erbB2 inhibition (and erbB2 expression in the case of the human lines) and also role of erbB2 via an alternative approach.

9) Fig 4a. These data are very odd and suggest that the effect of the inhibitor is not via erbB2. I assume that the experiment uses mubritinib, but its not specified. The inhibitor should achieve significant inhibition of erbB2 phosphorylation within minutes. However the effect at 24 h is not clear and a 2 fold inhibition is only achieved at 72 h. The effects on pErk and pAkt are also modest. Similar data are shown in Ext data 3 (note this figure shows signalling data, not viability data, as described in the text).

10) In 4b and Ext data 3b the effects on cGas and pSting are very modest and variable.

The authors state that 'These results indicate that ERBB2 inhibits TBK1 and cGAS primarily via AKT and RIG-I via MAP kinase signaling' but in Ext data 3a there is no effect of the erbB2i on pErk at 24 h but RIG-I is significantly increased.

11) Fig 5 is poorly described (the relevance of the pie chart is unclear) and not convincing. There is not a clear inverse correlation between erbB2 and MHC-1. If the authors have analysed a cohort as they suggest, then they should be able to provide better quantitative data to support this assertion.

12) Fig 6b provide quantitative data for metastasis.

13) Fig 9g the effect of PD-1 inhibitor on MHC-1 is not significant. Why are these data so different from data in Ext data 5c? Also, panels should be referred to in text in the correct order.

13) Fig 9e show data for all treatment groups.

14) Fig 9f-h, the data should be considered and described in more detail. Although erbB2i treatment increases MHC-1 and the number of activated CD4 and CD8 positive cells, the data for these parameters for combined treatment are very similar to PD-1 inhibitor alone. Notably the statistical analyses do not compare all relevant treatment groups. This impacts interpretation of the experiment.

Also Ext data 5a-b do not appear to be referred to in the text. Ext data 5b shows that anti-PD1 significantly reduces erbB2 expression. This makes the entire experiment difficult to interpret.

Reviewer #3

(Remarks to the Author)

metastatic tumours in a broad range of organ sites, including the liver. Despite, exhibiting a high tumour mutational burden, the recent addition of immune checkpoint blockage (ICB) therapy in combination with platinum-based chemotherapy fails to elicit durable responses in most SCLC patients. Therefore, understanding the molecular mechanisms that allow SCLC to

evade immune detection may lead to novel therapeutic approaches that enhance anti-tumour immunity in small cell lung cancer (SCLC). To gain new insights into the processes governing immune evasion in metastatic SCLC cells, Meder and colleagues interrogate paired tumour samples from SCLC patients and genetically engineered mouse (GEM) models using proteomic (e.g., IHC and phospho-proteomics) approaches. Specifically, MHC-I expression was lower in liver metastasis compared to matched lung SCLC tumours (human + mouse), with KO of MHC-I in murine SCLC resulting in enhanced metastatic dissemination, suggesting that MHC-I expression is a key mediator of anti-tumour immunity in SCLC. Phospho-proteomics identified the differential expression of ERBB2 in primary vs liver metastasis, suggesting that ERBB2 signalling may control MHC-I expression. Indeed, MHC-I expression (and components of the antigen presentation pathway) were induced in murine SCLC cells harbouring KO of ERBB2. This phenotype could be replicated upon chemical inhibition of ERBB2 in SCLC cells, although an effect is only shown when ERBB2i is combined with IFN γ stimulation. Importantly, ERBB2 inhibition synergized with ICB, highlighting a novel combination treatment strategy to enhance treatment response in SCLC patients.

This is an interesting study addressing a critical unmet need in SCLC patients. What is lacking however, is discussion of prior work describing the epigenetic regulation of MHC-I expression in SCLC. It would be important for the authors to discuss their findings in the context of this work. One major gap of this work is the poor citation and discussion of these key studies. It was also unclear the size of patient cohorts initially used for the investigation and for validation studies. It would appear that sample sizes are low, and thus these would need to be increased to make the findings more robust. Major gaps would need to be filled for this work to be suitable for publication in Nature Communications.

Comments:

1. It is unclear how many SCLC patients with 'matched' primary lung tissue and metastatic tissue were examined and formed part of the "Cologne SCLC Cohort". Numbers of patients examined should be included in the schematic (Fig. 1a).
2. In relation to the study approach, was MHC-I expression the sole biomarker interrogated by IHC staining in the paired patient samples? Or did the authors also examine other immune modulatory proteins?
3. In Fig. 1b-c, the authors show representative IHC images of MHC-I expression in lung and liver tumours in two representative patients. In the images shown, it is hard to discern whether membrane staining is seen. Moreover, further clarification of how 'relative MHC-I expression' is calculated would be beneficial and quantification of H-scores for each sample would allow the comparison of MHC-I staining patterns with prior studies (e.g., Mahadevan et al. *Cancer Discovery* 2021, PMID: 33707236) to be performed.
4. In the RP GEM model, MHC-I expression is examined in paired lung and liver tumours isolated from individual mice. CD56 expression is used to gate on SCLC tumour cells. The SCLC tumours that develop in RP mice, however, have been shown to exhibit neuroendocrine (NE) high (CD56+) and non-neuroendocrine (non-NE) CD56low/neg/CD44+ cells, with the latter in previous studies being the SCLC cell population that displays high MHC-I expression. What is the MFI of MHC-I expression across all murine SCLC samples examined? Are there any CD56low/negative tumour cells that show MHC-I-high expression?
5. One caveat of this study is that it appears many, if not all syngeneic cell line experiments (e.g., Fig. 2) are performed using one RP-derived cell line. To enhance the robustness of the results presented, *in vivo* results should ideally be repeated with an independent RP-generated syngeneic cell line.
6. If the effect of enhanced metastatic spread due to a lack of immune cell control, do MHC-I KO cells seed liver metastasis to a similar extent to WT cells, when IV injected into immune-deficient NSG or CD8 T cell-deficient mice. It would be interesting to see whether the effect was CD8 T or NK cell specific, given prior studies have shown that NK cells play a key role in controlling the metastatic spread of SCLC tumour cells (e.g., Best et al. *Journal of Thoracic Oncology* 2020; PMID: 32470639; Guanizo et al., *Nature Immunology* 2024; PMID: 39572642).
7. Several immune cell populations were evaluated in metastatic livers – this data should also be examined relative to tumour nodules. Do WT metastatic lesions exhibit more infiltrating immune cells?
8. Were metastatic lesions observed in additional tissue sites in mice IV injected with MHC-I KO cells? Other frequent metastatic sites include kidney, adrenal gland and bone.
9. If ERBB2 is controlling MHC-I expression and this function is important for controlling anti-tumour immunity:
 - a. Does IV injection of ERBB2 KO cells abolish liver metastasis?
 - b. Do ERBB2 KO cells exhibit similar *in vivo* growth (survival) kinetics to WT control cells, when injected into immune-deficient NSG mice?
 - c. Prior studies have also shown that re-expression of MHC-I expression in SCLC induces a non-NE phenotype. Did the authors observe a phenotypic change in the murine SCLC upon ERBB2 KO and/or ERBB2i?
10. Fig. 3h is mislabelled. Histograms represent MHC-I expression and not HER2 expression.
11. An ERBB2 inhibitor in combination with IFN γ stimulation in Fig. 3f demonstrated a further increase in MHC-I expression, compared to IFN γ stimulation alone. Given that the authors show that genetic loss of ERBB2 alone, results in increased MHC-I expression, does ERBB2i alone also enhance basal expression levels of MHC-I in SCLC cells? This should also be examined in the context of dual ERBB2 and STING inhibition (Fig. 4d).
12. Unclear in Fig. 5 if the pie chart is only representative of n=3 patient samples. If so, a larger patient cohort should be examined. Is higher ERBB2 expression seen in metastatic tumour lesions vs primary lung tumours, given that the authors show that paired metastatic lesions are in general low/negative for MHC-I (Fig. 1). Can the authors also interrogate publicly available datasets (including George et al. *Nature* 2015; PMID: 26168399 and Nabet et al. *Cancer Cell* 2024, PMID: 38366589) or human SCLC cell lines to further correlate ERBB2 vs MHC-I expression in patient samples.
13. Is the anti-tumour effect of ERBB2 inhibition (Fig. 6) mediated by CD8 T cells or other cytotoxic immune cell subsets? Immune cell depletion combined with transplantation models should be explored.
14. Does ERBB2 expression correlate with improved survival in SCLC patients treated in the Impower133 clinical trial? If ERBB2 is controlling MHC-I expression, then the authors might predict that ERBB2 expression correlates with patients that gained benefit from chemotherapy + ICB (Nabet et al. *Cancer Cell* 2024; PMID: 38366589).

15. Referencing of key publications in the field, particularly prior studies that have looked at regulation of MHC-I expression in SCLC. These papers are a key omission and should be referenced and discussed within the context of the findings of this work.

16. Some methods that describe how data was examined – particularly IHC quantifications – was lacking.

Version 1:

Reviewer comments:

Reviewer #1

(Remarks to the Author)

The questions raised during initial review have been largely addressed.

Reviewer #2

(Remarks to the Author)

The authors have addressed the majority of my concerns in a satisfactory manner.

The delayed effect of the ERBB2i on pathway activation is still unexpected and should at least be commented on in the text. In addition the authors indicate, correctly, that the T cell infiltration indicated by RNAseq data probably gives a more accurate representation of what is happening during the immune response, and this should also be discussed.

Minor points

- I don't think Ext data 1b is mentioned in the text
- Submitted Ext data tables 3 and 4 were identical
- There is still a tendency not to describe figure panels in the order they are presented in the figure eg in Fig 3
- Line 245, increase in RIG-I is Fig 3g
- Line 262, effect of ERBB2 therapeutic inhibition on MHC-1 is Fig 3k-i
- Ext data 4b, quantify Ki67
- Fig 9f - up regulation of MHC-1 with ERBB2i - this isn't significant, its a trend

Reviewer #3

(Remarks to the Author)

I would like to firstly thank the authors for taking on feedback and conducting additional experiments to address the reviewers concerns. The additional experiments and edits made to the manuscript have improved the quality of the work presented.

I have some (minor) comments that I would like addressed prior to publication in Nature Communications. I have outlined these below:

(1) The wording of the below statements require further consideration to enhance readability:

- In the abstract - "SCLC cells finally acquire the ability to evade immunosurveillance and resistance against immune checkpoint blockade"
- Lines 160-162: "In line, neuroendocrine SCLC with high T- effector signals demonstrated longer overall survival with PD-1 blockade and combined chemotherapy."

(2) I feel that the message (conclusion) of the manuscript should be toned down, as the data presented in this manuscript is not the only mechanism presented to underly the low expression of MHC-I in SCLC. The authors should consider modifying the following statement (lines 164-166): "Our data provide a mechanistic explanation for how SCLC loses MHC-I expression enabling immune cell evasion and strong evidence that combined inhibition of ERBB2 and PD-1 improves outcomes for SCLC patients."

(3) I was not aware that the Chan paper contained data that included response to ICB.

Statement (lines 296-299): "In line with our findings, the analysis of publicly available scRNA-seq datasets revealed a comparable frequency of patients with high ERBB2 expression samples in 26.37%. Moreover, high ERBB2 expression exhibited a trend toward shorter overall survival in SCLC patients treated with immune checkpoint therapy in combination with chemotherapy"

- Bulk RNA-sequencing data is available (upon request via the EGA) for n=271 SCLC patients enrolled in the The IMpower133 clinical trial that led to the FDA approval of anti-PDL1 + chemotherapy in SCLC patients. This dataset should be interrogated by the authors to enhance the robustness of the finding and conclusions drawn in this manuscript

Minor point:

(1) The morphology of the SCLC tumours transplanted into NSG mice does not appear concordant with SCLC. Staining tumours for pathological markers of SCLC, e.g., NCAM1 (CD56), INSM1 would be informative.

Point-by-point discussion of reviewer comments

ERBB2 signaling drives immune cell evasion and resistance against immunotherapy in small cell lung cancer

We thank you very much for reviewing our manuscript and the very valuable feedback of all three reviewers.

Reviewer 1

In this manuscript, Ullrich et al. investigate a causative role of ERBB2 signaling and MHC-I downregulation, which resulted in immune evasion of SCLC, a highly aggressive and treatment-resistant cancer. Accordingly, genetic or pharmacological blockade of ERBB2 restores MHC-I expression and T cell infiltration, overcomes immune evasion, and reduces SCLC metastasis. Furthermore, combining ERBB2 inhibition with PD-1 blockade elicits promising therapeutic responses in preclinical SCLC models.

Overall, the manuscript addresses an interesting topic in cancer immunology and presents an immune evasion mechanism in SCLC with translational potential. The data are generally convincing and support the conclusions drawn. The manuscript is well-written and presents its findings clearly.

1) A more detailed mechanistic explanation of ERBB2 signaling suppresses MHC-I expression is recommended. For example, how does cGAS-STING signaling drive the expression of MHC-I in SCLC to achieve this effect? Is it a direct effect of type I interferons or an indirect impact of IFN-gamma?

Reply: To address this important question by the reviewer whether the STING mediated effect on expression of MHC-I is a effect of Type I interferons or IFN γ we treated SCLC cell lines with IFN β and IFN γ in the context of ERBB2 and STING inhibition. Our data indicate that STING inhibition is able to counteract the effect of Type I interferons and to a minor extent IFN γ regarding MHC-I upregulation (new Ext. data Fig. 5c-e, lines 287-290).

2) How is ERBB2 signaling upregulated in these metastatic SCLC specimens through gene amplification, mutation, overexpression, or augmented by overexpression of EGFR, Notch, or Wnt?

Reply: We thank the reviewer for this insightful question regarding potential mechanisms of ERBB2 signaling in metastatic SCLC. To address this point, we performed whole-exome sequencing (WES) on SCLC cell lines derived from primary tumors and metastatic lesions. Importantly, we did not detect mutations in ERBB, EGFR, NOTCH, or WNT pathway genes

(new Ext. Data Tables 1+2; new Ext. Data Fig. 2). Furthermore, ERBB2 gene amplification was assessed by copy number analysis (new Ext. Data Fig. 2c). To investigate potential transcriptional regulation, we analyzed the transcriptomes of murine SCLC cell lines derived from primary and metastatic sites. We found no significant differential expression in genes associated with the EGFR, WNT, or NOTCH signaling pathways (new Ext. Data Fig. 2a). Additionally, we assessed publicly available single-cell RNA-seq data from human primary and metastatic SCLC and evaluated gene sets related to EGFR, WNT, and NOTCH signaling. Consistent with our murine data, we observed no differential upregulation of WNT or NOTCH pathway genes in metastatic samples (new Ext. Data Fig. 2b, lines 210-217).

Reviewer 2

In this manuscript the authors address the need for improved targeted treatment strategies against SCLC. They report decreased expression of MHC-1 in metastatic SCLC using patient samples and that this also occurs in a Rb1/Trp53-depleted SCLC mouse model. Use of the same model confirmed a functional role for MHC-1 in regulating metastasis and identified erbB2 as a mediator of MHC-1 repression. Further work then characterizes the mechanism of erbB2 regulation of MHC-1 and evaluates the potential of anti-erbB2 + anti-PD1 therapy. Although the work is interesting and addresses an important clinical question, the impact of the manuscript is limited by several factors. First, although I understand the limited number of syngeneic models of SCLC, there is an over-reliance on use of the original autochthonous mouse model, and use of cells from related mouse models with additional modifications would strengthen their conclusions. Second, the data from human patients of an inverse relationship between erbB2 and MHC-1 is not convincing. Third, there are concerns regarding the selectivity and effects of the erbB2i, as detailed below. Finally, some of the experiments are poorly described and lack experimental detail, and I'm not convinced by the interpretation of certain experiments, also explained below.

1) In Methods for 1b-c it states that clinico-pathological characteristics have been collected for the patient specimens but these are not provided.

Reply: As suggested by the reviewer, we added clinico-pathological characteristics of patients (new Ext. Data Table 10) serving for the matched and the new non-matched samples (new Fig. 5a-d) in the revised manuscript.

2) In 2a-c, metastasis is monitored following intravenous injection. This is not the same as metastasis from the primary site because it skips initial tumour invasion and intravasation into the bloodstream. In the orthotopic model (Extended Data Fig.1a-d) they just look at primary

tumour growth. What happens if the primary tumours are resected at the same size and metastasis is measured?

Reply: We thank the reviewer for raising this important point. To address this comment, we performed orthotopic injections and analyzed different metastatic sites. FACS analysis reveals the presence of metastatic tumor cells in the liver (new Ext. Data Fig. 1c-f, lines 193-197). However, large established metastatic lesions were not detected by IHC, likely due to the rapid progression of the primary tumor and early mortality in this orthotopic model, which restricts the time available for metastatic outgrowth.

3) Fig 3. It is unclear how many cell lines are being measured, and the identify/nature of these lines. In the legend it states n=4 and n=5 but it is unclear if this is referring to no of cell lines, or biological replicates. If no of cell lines, are these from different mice, or were multiple lines derived from the same tumour?

Reply: We apologize for the inappropriate annotations. We have now clarified that we applied a SCLC cell line that we established from a primary murine SCLC tumor and another SCLC cell line ex vivo cultured from a metastatic lesion. For both cell lines we performed 3 replicates. We added this information in the figure legend (lines 789-790).

I assume Figs 3a,c,d provide important data but insufficient info is provided to correctly interpret them.

Reply: We performed novel additional proteomics data to provide a higher accuracy of the proteomic data and to extend the sensitivity for the analysis of additional pathways and analyzed the samples with a higher protein quantity and a more sensitive detection method (new Fig. 3, lines 828-829). As suggested by the reviewer we here added more detailed information on proteomic analysis (line 604-657).

In 3c its unclear what the relevance of overexpressed MAPK1 and MAPK3 is if the phosphorylated form is not enhanced (3d).

Reply: To address this reviewer's important point we performed additional Western blot analysis and found increased pERK 1/2 (Thr202/Tyr204) in metastatic lesions in comparison to primary SCLC lesions (see new Ext. data Fig. 3l, lines 204).

Also the MAPKs that are upregulated in the metastatic samples are p38 stress-activated MAPKs ie MAPK13. These are not 'classical' erbB2 effectors and the authors should indicate this.

Reply: We thank the reviewer for this valuable comment and focused on downstream effectors of ERBB2 signaling. We changed the Figure and paragraph accordingly (new Fig. 3, new Ext. data Fig. 3).

In 3d the specific regulated phospho-sites are not indicated. This information must be provided because it is unclear whether the changed phosphorylation is actually on important regulatory sites.

Reply: We thank the reviewer for this important remark and added the phosphosites and their potential relevance in the Ext. Data Table 3+4.

4) MS methods are not provided beyond sample prep. Access to raw files is not provided.

Reply: We added a detailed description of the MS methods and uploaded the raw files (lines 604-657). Proteomic data are available in the PRIDE database (accession no. PXD065735) and in the MassIVE repository (accession no. MSV000098579). We updated the data availability statement, accordingly (line 735-744).

5) Fig 3f. Where is confirmation that knock-out did not affect SCLC proliferation?

Again, does it indicate independent knock-out lines, or replicates for one line?

Given that the lines are clonal, the authors should confirm that erbB2 rescue reverses the effects on gene expression.

Reply: To address this important comment, we compared ERBB2 KO cells vs ERBB2 WT SCLC cells and observed no significant changes in the proliferation rate in vitro and in vivo (new Ext. Data Fig. 4a, c, lines 309-310, 239-242). Additionally, we performed KI-67 IHC of these tumors and found no differences in tumor cell proliferation upon ERBB2 KO (new Ext. Data Fig. 4b). To strengthen these findings, we further performed cell titer glow experiments that also indicated no reduced cell viability upon ERBB2 inhibition (new Ext. data Fig. 4e). In the revised figure we annotated the used cell lines to each heatmap and graph accordingly. Finally, we rescued ERBB2 expression in two independent ERBB2 KO clones and could confirm that rescued ERBB2 expression downregulates MHC-I related gene expression (B2M and H2-D1) (new main Fig. 3e, new Ext. data Fig. 5h, lines 231-232).

6) 3h why is FACS plot labelled HER2, should be MHC-1

Reply: We apologize for the mislabeling and corrected accordingly (Fig. 3h).

7) Fig 3i, poorly explained. All legend says is 'mean protein expression scores'. How was this experiment done? What is the relevance of phosphorylation of these proteins? Also, these experiments utilized mubritinib as an erbB2i, but this drug also targets the electron transport chain. The authors should confirm mubritinib-induced inhibition of erbB2 in the cells and utilize another approach to confirm that the effect is erbB2-dependent eg knockdown.

Reply: We thank the reviewer for this remark and now provide an in-detail description of the proteomic analysis (lines 604-657) and focused on ERBB2 downstream effectors. To address this important point, we treated SCLC cells with the ERBB2 inhibitors lapatinib and neratinib that both do not affect electron transport chains. In concordance to mubritinib, we found an upregulation of MHC-I upon treatment with lapatinib and neratinib in human and murine SCLC cell lines (new Ext. Data Fig. 5a, b, lines 263-266). To further address this important point raised by the reviewer, we treated SCLC mice with lapatinib in combination with anti-PD1 and could validate our in vivo findings that combined ERBB2 and anti-PD1 treatment strongly improves anti-PD1 treatment efficacy (new Ext. Data Fig. 5i, lines 386-388). To further exclude a potential impact of the electron transport chain, we treated SCLC cells with a ERBB2-KO with mubritinib. Here, we did not observe a change in MHC-I expression assuming that the effect is ERBB2 dependent and independent of the electron transport chain (new Ext. Data Fig. 5f).

8) Fig 3j-k. Again, confirm erbB2 inhibition (and erbB2 expression in the case of the human lines) and also role of erbB2 via an alternative approach.

Reply: To address this important point, we treated SCLC cells with the alternative ERBB2 inhibitors lapatinib and neratinib in human and murine cell lines (new Ext. Data Fig. 5a, b). Consistent with our observations, alternative ERBB2 inhibitors as neratinib and lapatinib treatment led to an upregulation of MHC-I expression (new Ext. Data Fig. 5a, b, lines 263-266). Moreover, in an in vivo model, we tested combined treatment with lapatinib and anti-PD-1. As mentioned above, we could observe a significantly prolonged survival, similar to the therapeutic benefit observed with mubritinib plus anti-PD-1 therapy (new Ext. Data Fig. 5i, lines 386-388). Moreover, we could confirm ERBB2 expression in human cell lines with increasing ERBB2 expression in metastasis (Fig. 3b).

9) Fig 4a. These data are very odd and suggest that the effect of the inhibitor is not via erbB2. I assume that the experiment uses mubritinib, but its not specified. The inhibitor should achieve significant inhibition of erbB2 phosphorylation within minutes. However the effect at 24 h is not clear and a 2 fold inhibition is only achieved at 72 h. The effects on pErk and pAkt are also modest. Similar data are shown in Ext data 3 (note this figure shows signalling data, not viability data, as described in the text).

Reply: To address the reviewer's remark we performed WB analysis to earlier time points: After 10min, 30min, 2h and 4h. We did not observe a decrease in pERK or pAKT at these early time points of ERBB2 inhibition (new Ext. Data Fig. 9e). We hypothesize that the delayed decline is related to the fact that ERBB2 is not a dominant driver of cell survival in SCLC in contrast to e.g. constitutively activated EGFR where we would expect and observe inhibition of downstream signaling after minutes or a few hours. We further clarified that the figure shows signaling data in the text (lines 275-277)

10) In 4b and Ext data 3b the effects on cGas and pSting are very modest and variable. The authors state that 'These results indicate that ERBB2 inhibits TBK1 and cGAS primarily via AKT and RIG-I via MAP kinase signaling' but in Ext data 3a there is no effect of the erBB2i on pErk at 24 h but RIG-I is significantly increased.

Reply: We agree with the reviewer that the impact of AKTi on cGAS and pSTING is rather modest. We therefore modified this interpretation of these data and changed the paragraph accordingly (lines 283-286).

11) Fig 5 is poorly described (the relevance of the pie chart is unclear) and not convincing. There is not a clear inverse correlation between erBB2 and MHC-1. If the authors have analysed a cohort as they suggest, then they should be able to provide better quantitative data to support this assertion.

Reply: To address the reviewer's insightful comment, we performed a quantitative immunohistochemical analysis of ERBB2 and MHC-I expression in our cohort of human SCLC tumor samples using QuPath-based image analysis. Here, we could show an inverse correlation between ERBB2 and MHC-I ($r=-0,45$; $p=0.01$) (new Fig. 5d). We further corroborated these findings by analyzing an independent single-cell RNA sequencing dataset derived from primary and metastatic SCLC patient samples. This analysis revealed a consistent loss of MHC-I and STING, alongside with increased ERBB2 and AKT pathway signaling in metastatic lesions compared to primary tumors (new Fig. 5f) (lines 291-302).

12) Fig 6b provide quantitative data for metastasis.

Reply: To address the reviewer's comment, we have included quantitative data for metastasis in Fig. 6b (lines 308-309). Our findings show a higher metastatic infiltration into the liver in the vehicle group, which was significantly reduced following ERBB2 inhibition.

13) Fig 9g the effect of PD-1 inhibitor on MHC-1 is not significant. Why are these data so different from data in Ext data 5c? Also, panels should be referred to in text in the correct order.

Reply: We would like to thank the reviewer for the attentive comment. We extended the mouse treatment cohort and included additional mice to increase the sample size. With the increased sample size we can now provide significant evidence that treatment with anti-PD-1 reduces MHC-I expression (new Fig. 9f, lines 372-375). To corroborate these findings, we analyzed scRNA-seq data from SCLC patients treated with immune checkpoint inhibitors and could confirm that immune checkpoint targeted therapy mediates loss of B2M on SCLC cells (new Fig. 9h, lines 396-399).

14) Fig 9e show data for all treatment groups.

Reply: As suggested, we added the pie charts of vehicle and ERBB2i treated SCLC mice (new Fig. 5e).

15) Fig 9f-h, the data should be considered and described in more detail. Although erbB2i treatment increases MHC-1 and the number of activated CD4 and CD8 positive cells, the data for these parameters for combined treatment are very similar to PD-1 inhibitor alone. Notably the statistical analyses do not compare all relevant treatment groups. This impacts interpretation of the experiment.

Also Ext data 5a-b do not appear to be referred to in the text. Ext data 5b shows that anti-PD1 significantly reduces erbB2 expression. This makes the entire experiment difficult to interpret.

Reply: We thank the reviewer for this important remark and described the data in more detail (lines 394-394). We also added statistical analyses between all treatment groups confirming that ERBB2 inhibition significantly increases the infiltration of activated CD4 and CD8 cells (new Ext. data Fig. 11c, d). We acknowledge that the levels of activated CD4⁺ and CD8⁺ T cells in the combination therapy group appear similar to those observed in the anti-PD-1 treatment group based on the FACS analysis. However, it is important to note that the FACS analysis was performed at the end of the experiment, when the mice were in an advanced stage of disease progression. Consequently, the immune compartment at this time point may not fully reflect the initial immune response to therapy. In contrast, the scRNA-seq data (Fig. 7), which was generated from samples collected during the response, provides a more accurate representation of the immune landscape. This analysis revealed a significantly higher infiltration of T cells, particularly CD8⁺ T cells, in the combination therapy group compared to the anti-PD-1 monotherapy group. We hope this clarification addresses the concerns raised and facilitates a more accurate interpretation of our findings.

Reviewer 3

metastatic tumours in a broad range of organ sites, including the liver. Despite, exhibiting a high tumour mutational burden, the recent addition of immune checkpoint blockage (ICB)

therapy in combination with platinum-based chemotherapy fails to elicit durable responses in most SCLC patients. Therefore, understanding the molecular mechanisms that allow SCLC to evade immune detection may lead to novel therapeutic approaches that enhance anti-tumour immunity in small cell lung cancer (SCLC). To gain new insights into the processes governing immune evasion in metastatic SCLC cells, Meder and colleagues interrogate paired tumour samples from SCLC patients and genetically engineered mouse (GEM) models using proteomic (e.g., IHC and phospho-proteomics) approaches. Specifically, MHC-I expression was lower in liver metastasis compared to matched lung SCLC tumours (human + mouse), with KO of MHC-I in murine SCLC resulting in enhanced metastatic dissemination, suggesting that MHC-I expression is a key mediator of anti-tumour immunity in SCLC. Phospho-proteomics identified the differential expression of ERBB2 in primary vs liver metastasis, suggesting that ERBB2 signalling may control MHC-I expression. Indeed, MHC-I expression (and components of the antigen presentation pathway) were induced in murine SCLC cells harbouring KO of ERBB2. This phenotype could be replicated upon chemical inhibition of ERBB2 in SCLC cells, although an effect is only shown when ERBB2i is combined with IFN γ stimulation. Importantly, ERBB2 inhibition synergized with ICB, highlighting a novel combination treatment strategy to enhance treatment response in SCLC patients.

This is an interesting study addressing a critical unmet need in SCLC patients. What is lacking however, is discussion of prior work describing the epigenetic regulation of MHC-I expression in SCLC. It would be important for the authors to discuss their findings in the context of this work. One major gap of this work is the poor citation and discussion of these key studies. It was also unclear the size of patient cohorts initially used for the investigation and for validation studies. It would appear that sample sizes are low, and thus these would need to be increased to make the findings more robust. Major gaps would need to be filled for this work to be suitable for publication in Nature Communications.

1) It is unclear how many SCLC patients with 'matched' primary lung tissue and metastatic tissue were examined and formed part of the "Cologne SCLC Cohort". Numbers of patients examined should be included in the schematic (Fig. 1a).

Reply: As suggested, we added and annotated the number of patients included in the SCLC patient cohort (see adapted Fig. 1a).

2) In relation to the study approach, was MHC-I expression the sole biomarker interrogated by IHC staining in the paired patient samples? Or did the authors also examine other immune modulatory proteins?

Reply: We thank the reviewer for this question. In the paired patient samples, we only examined MHC-I expression by IHC. Due to the limited amount of available tissue, it was not feasible to perform additional staining for other immune modulatory proteins.

3) In Fig. 1b-c, the authors show representative IHC images of MHC-I expression in lung and liver tumours in two representative patients. In the images shown, it is hard to discern whether membrane staining is seen. Moreover, further clarification of how 'relative MHC-I expression' is calculated would be beneficial and quantification of H-scores for each sample would allow the comparison of MHC-I staining patterns with prior studies (e.g., Mahadevan et al. *Cancer Discovery* 2021, PMID: 33707236) to be performed.

Reply: We thank the reviewer for these helpful comments. The relative MHC-I expression presented was calculated as the area of MHC-I positive cells divided by the total tumor area. However, in response to the reviewer's suggestion and to allow better comparability with prior studies (e.g., Mahadevan et al., *Cancer Discovery* 2021, PMID: 33707236), we have now additionally calculated and provided the H-scores for each sample. We could herewith validate our findings demonstrating a loss of MHC-I in metastatic lesions (new Fig. 1e, lines 176-180).

4) In the RP GEM model, MHC-I expression is examined in paired lung and liver tumours isolated from individual mice. CD56 expression is used to gate on SCLC tumour cells. The SCLC tumours that develop in RP mice, however, have been shown to exhibit neuroendocrine (NE) high (CD56+) and non-neuroendocrine (non-NE) CD56low/neg/CD44+ cells, with the latter in previous studies being the SCLC cell population that displays high MHC-I expression. What is the MFI of MHC-I expression across all murine SCLC samples examined? Are there any CD56low/negative tumour cells that show MHC-Ihigh expression?

Reply: In response to the reviewer's comment, we assessed the MFI of MHC-I expression across murine SCLC samples and found no significant differences between CD56low/negative and CD56high/positive tumor cells (new Ext. data Fig. 8d). To further investigate this, we analyzed scRNA-seq data from primary murine SCLC samples across all therapy groups. Here we found significant negative correlation between neuroendocrine markers (NE score) and MHC-I (new Ext. data Fig. 8c). Moreover, we also detected a negative correlation of neuroendocrine marker genes with CD44, which is in line with current literature (new Ext. data Fig. 8b, new references 29 and 30, see discussion 425-430). We added these new findings in the manuscript (lines 260-263).

5) One caveat of this study is that it appears many, if not all syngeneic cell line experiments (e.g., Fig. 2) are performed using one RP-derived cell line. To enhance the robustness of the

results presented, in vivo results should ideally be repeated with an independent RP-generated syngeneic cell line.

Reply: As recommended, we performed additional in vivo experiments. We here could validate our initial findings with an additional murine SCLC (see new Ext. Data. Fig. 4, 6 and 7)

6) If the effect of enhanced metastatic spread due to a lack of immune cell control, do MHC-I KO cells seed liver metastasis to a similar extent to WT cells, when IV injected into immune-deficient NSG or CD8 T cell-deficient mice. It would be interesting to see whether the effect was CD8 T or NK cell specific, given prior studies have shown that NK cells play a key role in controlling the metastatic spread of SCLC tumour cells (e.g., Best et al. *Journal of Thoracic Oncology* 2020; PMID: 32470639; Guanizo et al., *Nature Immunology* 2024; PMID: 39572642).

Reply: To address this important question, whether the enhanced metastatic formation of MHC-I KO SCLC is due to reduced CD8⁺ T cell-mediated tumor control, we performed additional in vivo experiments. We intravenously injected SCLC MHC-I KO and SCLC WT cells into NSG mice and observed comparable metastatic engraftment in both groups (new Fig. 2h). To further dissect the role of immune cell subsets, we conducted in vivo CD4⁺/CD8⁺ T cell depletion experiments. Here, we found that the depletion of CD4⁺ and CD8⁺ T cells led to increased tumor growth of ERBB2 KO cells, despite the presence of functional NK cells (new Ext. data Fig. 7a, b). These data suggests that immune surveillance in our model is predominantly mediated by adaptive immunity, and that NK cells alone are insufficient for tumor control. We addressed this important finding in the manuscript (lines 237-242).

7) Several immune cell populations were evaluated in metastatic livers – this data should also be examined relative to tumour nodules. Do WT metastatic lesions exhibit more infiltrating immune cells?

Reply: To address this important point, we assessed immune cell populations in metastatic livers relative to the tumour nodules. We found that WT metastatic lesions exhibit more infiltrating immune cells particularly T cells in comparison to MHC-I KO nodules (new Fig. 2f). Moreover, we found higher infiltration of activated CD8 T cells in MHC-I WT tumors in comparison to MHC-I KO tumors (new Fig. 2g, lines 188-191).

8) Were metastatic lesions observed in additional tissue sites in mice IV injected with MHC-I KO cells? Other frequent metastatic sites include kidney, adrenal gland and bone.

Reply: We thank the reviewer for this valuable remark. To address this point, we further analyzed potential metastatic sites following intravenous injection of MHC-I KO SCLC cells into immunocompetent mice. In these experiments, we observed metastatic spread to the liver

and lymph nodes. However, we did not detect metastases in the brain, bones, pancreas, spleen, or kidneys (new Ext. data Fig. 1a).

9) If ERBB2 is controlling MHC-I expression and this function is important for controlling anti-tumour immunity:

a) Does IV injection of ERBB2 KO cells abolish liver metastasis?

Reply: To address this comment, we performed additional mice experiments and injected ERBB2 KO SCLC intravenously into immunocompetent mice. Here, we found a strong reduction of liver metastasis in comparison to ERBB2 WT SCLC cells. Additionally, when ERBB2 KO SCLC cells were injected orthotopically mice showed a prolonged survival in comparison to the WT, indicating better tumor control via antigen –presentation of the ERBB2 KO tumor cells (new Ext. data Fig. 6, lines 232-237).

b) Do ERBB2 KO cells exhibit similar in vivo growth (survival) kinetics to WT control cells, when injected into immune-deficient NSG mice?

Reply: To address this comment, we performed additional mice experiments and injected ERBB2 KO SCLC sc. into NSG mice. Here, we found no significant differences in growth kinetics between ERBB2 WT SCLC tumors and ERBB2 KO SCLC tumors (new Ext. data Fig. 4a, b, lines 239-242).

c) Prior studies have also shown that re-expression of MHC-I expression in SCLC induces a non-NE phenotype. Did the authors observe a phenotypic change in the murine SCLC upon ERBB2 KO and/or ERBB2i?

Reply: We thank the reviewer for these valuable suggestions. To address this point, we analysed a potential shift of the neuroendocrine phenotype upon ERBB2i. Here, we did not observe a significant shift of the neuroendocrine phenotype in vivo and in vitro indicating that ERBB2 blockade does not impair neuroendocrine differentiation (new Ext. data Fig. 8, lines 260-263).

10) Fig. 3h is mislabelled. Histograms represent MHC-I expression and not HER2 expression.

Reply: We apologize for mislabeling and corrected accordingly.

11) An ERBB2 inhibitor in combination with IFN γ stimulation in Fig. 3f demonstrated a further increase in MHC-I expression, compared to IFN γ stimulation alone. Given that the authors show that genetic loss of ERBB2 alone, results in increased MHC-I expression, does ERBB2i alone also enhance basal expression levels of MHC-I in SCLC cells? This should also be examined in the context of dual ERBB2 and STING inhibition (Fig. 4d).

Reply: To address these suggestions, we treated SCLC with three different ERBB2i alone and observed an increase of basal expression levels of MHC-I (new Ext. data Fig. 5a, b). We further examined STING inhibition in the context of ERBB2 blockade and found that STING inhibition abolishes MHC-I expression (new Ext. data Fig. 5c). We addressed these new data in lines 263-266.

12) Unclear in Fig. 5 if the pie chart is only representative of n=3 patient samples. If so, a larger patient cohort should be examined. Is higher ERBB2 expression seen in metastatic tumour lesions vs primary lung tumours, given that the authors show that paired metastatic lesions are in general low/negative for MHC-I (Fig. 1). Can the authors also interrogate publicly available datasets (including George et al. Nature 2015: PMID: 26168399 and Nabet et al. Cancer Cell 2024, PMID: 38366589) or human SCLC cell lines to further correlate ERBB2 vs MHC-I expression in patient samples.

Reply: We thank the reviewer for these important comments. To address these points, we further corroborate our findings with an additional published scRNA data set of primary and metastatic SCLC patients confirming an increased expression of MHC-I (Fig. 1 b, c, lines 171-176). We further found enhanced STING expression in primary SCLC tumors and increased expression of ERBB2 and AKT in metastatic SCLC (new Fig. 5f). Additionally, we analysed tissue samples of SCLC patients and confirmed an inverse correlation between ERBB2 and MHC-I expression (new Fig. 5d, lines 291-294). We further analysed RNA data from SCLC cell lines and confirmed an inverse correlation of ERBB2 and B2M (new Ext. data Fig. 10a).

13) Is the anti-tumour effect of ERBB2 inhibition (Fig. 6) mediated by CD8 T cells or other cytotoxic immune cell subsets? Immune cell depletion combined with transplantation models should be explored.

Reply: To address this important point, we performed additional in vivo depletion experiments in ERBB2 KO SCLC and investigated the impact of CD4, CD8 depletion on tumor growth after injection of ERBB2 KO SCLC cells. Here, CD4, CD8 depletion strongly enhanced tumor growth of ERBB2 KO tumors undermining that the anti-tumor effect is CD4/CD8 mediated (new Ext. data Fig. 7a,b). We discussed this important finding in the manuscript (lines 237-239).

14) Does ERBB2 expression correlate with improved survival in SCLC patients treated in the Impower133 clinical trial? If ERBB2 is controlling MHC-I expression, then the authors might predict that ERBB2 expression correlates with patients that gained benefit from chemotherapy + ICB (Nabet et al. Cancer Cell 2024; PMID: 38366589).

Reply: We thank the reviewer raising this point. We were able to access the data from Cancer Cell 2021 Chan JM, Rudin CM and could confirm in a small available patient cohort that within

this small cohort of patients, SCLC patients with higher ERBB2 expression tend to be associated with a longer overall survival after immunochemotherapy (new Ext. data Fig. 10b). However, this finding remains to be validated in a larger prospective clinical trial.

15) Referencing of key publications in the field, particularly prior studies that have looked at regulation of MHC-I expression in SCLC. These papers are a key omission and should be referenced and discussed within the context of the findings of this work.

Reply: We thank the reviewer for these valuable suggestions and discussed these key publications in the revised manuscript (lines 425-430).

16) Some methods that describe how data was examined – particularly IHC quantifications – was

Reply: As suggested, we in more detail described the methods applied in the manuscript in particular the description of the IHC methods (lines 500-504) lacking.

Point-by-point discussion of reviewer comments

We sincerely thank the reviewers for their time and effort in evaluating our revised manuscript. We greatly appreciate the constructive feedback provided during the review process, which has helped us to further improve our work.

Reviewer 1

The questions raised during initial review have been largely addressed.

Reply: We thank the reviewer for the valuable comments during the initial review and the positive assessment of our revision.

Reviewer 2

The authors have addressed the majority of my concerns in a satisfactory manner. The delayed effect of the ERBB2i on pathway activation is still unexpected and should at least be commented on in the text. In addition the authors indicate, correctly, that the T cell infiltration indicated by RNAseq data probably gives a more accurate representation of what is happening during the immune response, and this should also be discussed.

Reply: We thank the reviewer for this comment. We discuss the delayed effect of the ERBB2i on pathway activation in the revised manuscript (line 241-243). As recommended, we expanded our discussion of the T cell infiltration data and emphasize that scRNA-seq in response likely provides a more accurate representation of immune cell infiltration (line 317-320).

Minor points

I don't think Ext data 1b is mentioned in the text

Reply: We thank the reviewer for attentive remark. We addressed Ext data 1b in line 150 in the manuscript.

Submitted Ext data tables 3 and 4 were identical

Reply: We thank the reviewer for noticing this. We corrected the files and submit the correct versions of Extended Data Tables 3 and 4.

There is still a tendency not to describe figure panels in the order they are presented in the figure eg in Fig 3

Reply: We changed the order of the items in figure 3 accordingly.

Line 245, increase in RIG-I is Fig 3g

Reply: We thank the reviewer for the comment. The increase in RIG-I is now correctly indicated.

Line 262, effect of ERBB2 therapeutic inhibition on MHC-1 is Fig 3k-i

Reply: We thank the reviewer for careful reading. We now reference Figure 3k-I when describing the effect of ERBB2 therapeutic inhibition on MHC-I at line 262.

Ext data 4b, quantify Ki67

Reply: We thank the reviewer and will follow this suggestion. In the revised manuscript, we included a quantification of Ki67 (Supplementary Fig. 4b).

Fig 9f up regulation of MHC-1 with ERBB2i - this isn't significant, its a trend

Reply: We agree with the reviewer and changed the paragraph in the revised manuscript accordingly (lines 370-371).

Reviewer 3

I would like to firstly thank the authors for taking on feedback and conducting additional experiments to address the reviewers concerns. The additional experiments and edits made to the manuscript have improved the quality of the work presented. I have some (minor) comments that I would like addressed prior to publication in Nature Communications. I have outlined these below:

The wording of the below statements require further consideration to enhance readability:

- In the abstract - "SCLC cells finally acquire the ability to evade immunosurveillance and resistance against immune checkpoint blockade"
- Lines 160-162: "In line, neuroendocrine SCLC with high T- effector signals demonstrated longer overall survival with PD-L1 blockade and combined chemotherapy."

Reply: We thank the reviewer for this comment regarding clarity and readability. We revised the sentences accordingly to improve clarity.

I feel that the message (conclusion) of the manuscript should be toned down, as the data presented in this manuscript is not the only mechanism presented to underly the low expression of MHC-I in SCLC. The authors should consider modifying the following statement (lines "Our data provide a mechanistic explanation for how SCLC loses MHC-I expression enabling immune cell evasion and strong evidence that combined inhibition of ERBB2 and PD-1 improves outcomes for SCLC patients."

Reply: We agree with the reviewer's comment and as recommended revised the manuscript to present our findings with more balanced wording (lines 121-123).

I was not aware that the Chan paper contained data that included response to ICB. Statement (lines 296-299): "In line with our findings, the analysis of publicly available scRNA-seq datasets¹⁰ revealed a comparable frequency of patients with high ERBB2 in expression samples in 26.37%. Moreover, high ERBB2 expression exhibited a trend toward shorter overall survival in SCLC patients treated with immune checkpoint therapy in combination with chemotherapy".

Bulk RNA-sequencing data is available (upon request via the EGA) or n=271 SCLC patients enrolled in the IMpower133 clinical trial that lead the FDA approval of anti-PDL1 + chemotherapy in SCLC patients. This dataset should be interrogated by the authors to enhance the robustness of the finding and conclusions drawn in this manuscript

Reply: We thank the reviewer for highlighting the IMpower133 dataset. We requested the access to this data set and received the confirmation for access. We analyzed the data set and observed that high ERBB2 expression is associated with a shorter overall survival in the entire IMpower133 patient cohort. We also found a tendency of high ERBB2 expression and worse treatment outcome in the subgroup analysis of patients treated with the study arm Carboplatin/Etoposide/Atezolizumab though it was not significant.

As we identified ERBB2 as a relevant pathway that mediates the expression of MHC-I and thereby antigen presentation in SCLC, we further investigated the impact of ERBB2 expression in SCLC patients with high TMB. TMB has been shown to positively correlate with the number of neoantigens whereas its presentation is mediated by tumor intrinsic pathways (Holder et al., 2024) such as ERBB2 in our study. Corroborating our findings, we observed in the subgroup analysis of patients with high TMB, that high ERBB2 expression significantly correlates with worse outcome after combined Immune checkpoint inhibition with chemotherapy indicating that ERBB2 signaling contributes to presentation of neoantigens in SCLC patients. We feel that these findings strengthen

the impact of ERBB2 on outcome after treatment with ICB. We again thank this reviewer for this important remark and replaced Supplementary Fig. 10b with this new data and discussed these important data in the revised manuscript.

Fig. 1 for review: **a**, Overall survival of ERBB2 low vs. high patients in the IMpower133 trial cohort (n = 271). **b**, Overall survival of ERBB2 low vs. high patients in the carboplatin, etoposide + atezolizumab treatment arm of IMpower133 (n = 132). (new Supplementary Fig. 10b) **c**, Overall survival in the TMB-high subgroup of ERBB2 low vs. high patients in the IMpower133 trial cohort (n = 78). Expression cutoff was determined using Cutoff Finder (Budczies et al., 2012). Statistical analysis was done using the Mantel-Cox (log-rank) test (*, $p < 0.05$). (new Supplementary Fig. 10c)

The morphology of the SCLC tumours transplanted into NSG mice does not appear concordant with SCLC. Staining tumours for pathological markers of SCLC, e.g., NCAM1 (CD56), INSM1 would be informative.

Reply: To ensure the character of SCLC tumors in NSG mice, we will follow the reviewer's comment and stain for CD56, a typical neuroendocrine marker of SCLC. These analyses confirmed the expression of CD56 (Fig.2h, Supplementary Fig. 4b).

References

- [1] Holder AM, Dedeilia A, Sierra-Davidson K, et al. Defining clinically useful biomarkers of immune checkpoint inhibitors in solid tumours. *Nat Rev Cancer*. 2024;24(7):498-512. doi:10.1038/s41568-024-00705-7
- [2] Budczies J, Klauschen F, Sinn BV, et al. Cutoff Finder: a comprehensive and straightforward Web application enabling rapid biomarker cutoff optimization. *PLoS One*. 2012;7(12):e51862. doi:10.1371/journal.pone.0051862.